# LIVS: A Pluralistic Alignment Dataset for Inclusive Public Spaces

**Rashid Mushkani** [1 2]  **Shravan Nayak** [1 2]  **Hugo Berard** [1]  **Allison Cohen** [2]  **Shin Koseki** [1 2]  **Hadrien Bertrand** [2]

## Abstract

We introduce the *Local Intersectional Visual Spaces* (LIVS) dataset, a benchmark for multi-criteria alignment, developed through a two-year participatory process with 30 community organizations to support the pluralistic alignment of text-to-image (T2I) models in inclusive urban planning. The dataset encodes 37,710 pairwise comparisons across 13,462 images, structured along six criteria—Accessibility, Safety, Comfort, Invitingness, Inclusivity, and Diversity—derived from 634 community-defined concepts. Using Direct Preference Optimization (DPO), we fine-tune Stable Diffusion XL to reflect multi-criteria spatial preferences and evaluate the LIVS dataset and the fine-tuned model through four case studies: (1) DPO increases alignment with annotated preferences, particularly when annotation volume is high; (2) preference patterns vary across participant identities, underscoring the need for intersectional data; (3) human-authored prompts generate more distinctive visual outputs than LLM-generated ones, influencing annotation decisiveness; and (4) intersectional groups assign systematically different ratings across criteria, revealing the limitations of single-objective alignment. While DPO improves alignment under specific conditions, the prevalence of neutral ratings indicates that community values are heterogeneous and often ambiguous. LIVS provides a benchmark for developing T2I models that incorporate local, stakeholder-driven preferences, offering a foundation for context-aware alignment in spatial design.

⊕ https://mid-space.one

[1]Université de Montréal [2]Mila–Quebec AI Institute. Correspondence to: Rashid Mushkani <rashid.ahmad.mushkani@umontreal.ca>.

*Proceedings of the 42nd International Conference on Machine Learning*, Vancouver, Canada. PMLR 267, 2025. Copyright 2025 by the author(s).

## 1. Introduction

Recent advances in text-to-image (T2I) generative modeling have significantly improved image quality and diversity (Bai et al., 2022; Podell et al., 2023; Zhang et al., 2024a). These developments can benefit communities by making design processes more accessible—enabling broader engagement among non-expert stakeholders in architecture, urban planning, and environmental visualization. (Corner, 1999; Pallasmaa, 2011; Dubey et al., 2024). However, aligning T2I models with the specific needs of local communities remains an open challenge (Qadri et al., 2023; Kannen et al., 2024; Kirk et al., 2024), particularly when addressing subjective concepts such as *inclusivity* or *safety*.

Existing alignment frameworks often rely on large-scale, global data and crowdwork, which may not capture the nuanced objectives of smaller communities (Dzieza, 2023; Anthropic, 2023; OpenAI et al., 2024; Kirk et al., 2024). Moreover, T2I alignment research has often centered on broad aesthetic or content moderation goals (Kirstain et al., 2023; Pressman et al., 2022), while paying limited attention to diverse local criteria in domains where multiple social identities intersect. This gap poses a risk that generative models could systematically exclude or misrepresent historically marginalized groups in depictions of public spaces (Wan et al., 2024; Prerak, 2024).

To address these limitations, we propose a *pluralistic alignment* approach, wherein alignment explicitly accommodates multiple coexisting norms and values rather than seeking a single universal solution (Sorensen et al., 2024). We introduce the *Local Intersectional Visual Spaces* (LIVS) dataset, which integrates intersectional, community-driven feedback on T2I-generated images of urban public spaces. Over a two-year period, we collaborated with 30 community organizations through workshops and interviews, initially collecting 634 criteria for inclusive public space design. Through iterative co-creation, these were distilled into six broader categories: *Accessibility*, *Safety*, *Comfort*, *Invitingness*, *Inclusivity*, and *Diversity*.

We collected 35,510 multi-criteria preference annotations, each covering one to three criteria, to fine-tune a Stable Diffusion XL model using Direct Preference Optimization (DPO) (Wallace et al., 2023; Rafailov et al., 2024). We then tested the fine-tuned model with 2,200 additional an-

notations, comparing it to the baseline model. In these comparisons, 700 favored the DPO model, 300 favored the baseline, and about 1,100 were neutral, indicating that existing alignment methods are insufficient to capture diverse, subjective preferences and highlighting the need for more nuanced alignment approaches. Our experiments demonstrate that multi-criteria feedback can steer T2I models toward locally meaningful outputs, while also showing that no single alignment objective can account for the complexity of community priorities. Additional case studies demonstrate that preference patterns differ across participant identities, that human-authored prompts result in more visually distinct outputs than those generated by language models, and that intersectional groups provide systematically different ratings across criteria, further illustrating the limitations of single-objective alignment frameworks.

**Contributions:**

- We introduce the *Local Intersectional Visual Spaces* (LIVS) dataset, developed through a participatory methodology that captures diverse, community-generated dimensions of inclusive public space design (Berditchevskaia et al., 2021; Sloane et al., 2022).

- We propose a pluralistic alignment framework for text-to-image (T2I) generative models, focusing on intersectional and locally specific criteria within urban public space contexts. This framework underpins the construction of the LIVS dataset, tailored for the urban planning domain.

- We provide empirical evidence that DPO fine-tuning can modulate image generation according to multi-criteria feedback, yet the prevalence of neutral responses indicates the continued insufficiency of current alignment methods to capture complex intersectional preferences, underscoring the need for approaches capable of accommodating nuanced, diverse user needs. (Fan et al., 2023; Casper et al., 2023; Li et al., 2024; Rafailov et al., 2024).

- We demonstrate the influence of participant identities on model preferences and compare human-authored and AI-generated prompts, highlighting the importance of accommodating local pluralism and human creativity in T2I alignment.

We envision our approach bridging the gap between purely global alignment strategies and the need for more fine-grained methods that incorporate local, intersectional values in real-world applications. In the following sections, we contextualize related work, detail our methodology, describe our alignment experiments, and discuss broader implications for the field of machine learning.

## 2. Related Work

### 2.1. Alignment of Generative Models

Multiple datasets have supported alignment efforts in text-to-image (T2I) generation. For instance, Simulacra Aesthetic Captions (Pressman et al., 2022) offers 238,000 synthetic images rated for aesthetics, and Pic-a-Pic (Kirstain et al., 2023) comprises over 500,000 preference data points. ImageReward (Xu et al., 2023) extends these efforts by capturing ratings on alignment, fidelity, and harmlessness, while HPS (Wu et al., 2023) and HPS v2 (Wu et al., 2023) propose large-scale binary preference pairs to train reward models reflecting human judgments. Related work in language models has explored moral decision-making in multilingual contexts (Jin et al., 2024) and contextual preferences across diverse demographics (Kirk et al., 2024), highlighting the importance of subjective, multicultural perspectives in alignment processes.

Studies on T2I alignment often focus on aesthetic preferences (Pressman et al., 2022; Kirstain et al., 2023; Wu et al., 2023) or content policy compliance. However, they typically assume a single, global notion of "goodness" or "suitability." Pluralistic alignment, in contrast, recognizes that social values are heterogeneous and context-dependent (Turchin, 2019a; Sorensen et al., 2024). Our work extends beyond purely global alignment by collecting specialized, multi-criteria annotations grounded in local, intersectional community knowledge for urban public space design.

Fewer efforts target image-based generative models compared to alignment in large language models (Bai et al., 2022; Huang et al., 2024b). Prior research has primarily utilized reward modeling and reinforcement learning from human feedback (Stiennon et al., 2022; Ouyang et al., 2022), with a focus on single-objective tasks such as helpfulness or factual correctness (Kirk et al., 2024; Jin et al., 2024). In contrast, our approach leverages multi-criteria annotations to inform alignment in T2I outputs within an urban planning context, though these signals are ultimately aggregated into a single binary label for model training via majority voting.

### 2.2. Intersectionality and Local Knowledge

Intersectionality recognizes that individuals may experience multiple, overlapping forms of marginalization, affecting how they engage with public spaces and technology (Crenshaw, 1989; Costanza-Chock, 2020). In generative modeling, this perspective is often overlooked, with systems calibrated to an "average" user profile that can obscure the distinct needs of marginalized groups (Benjamin, 2019; Gebru et al., 2021; Kirk et al., 2024; Murgia, 2024). For urban planning tasks, ignoring intersectionality risks overlooking critical insights related to accessibility, safety, or cultural expression. Integrating local knowledge adds further granu-

larity, accounting for the historical, spatial, and communal context that global datasets typically lack (Fischer, 2000; Nekoto et al., 2020; Mohamed et al., 2020; D'Ignazio & Klein, 2020). While some studies acknowledge the importance of localized, context-specific input (Aroyo & Welty, 2015; Sloane, 2024; Sieber et al., 2024b), few systematically address intersectionality in T2I alignment. By weaving intersectional considerations into local community-driven annotations, our approach endeavors to reflect a broader range of perspectives and needs, moving beyond singular, one-size-fits-all criteria in generative modeling.

## 2.3. Visual Generative Modeling for Urban Spaces

Urban planning and design have long leveraged visualizations to communicate design objectives and gather feedback from stakeholders (Corner, 1999; Dubey et al., 2024; Guridi et al., 2024). T2I models offer the promise of more rapid prototyping and inclusive deliberation, especially when non-experts can directly prompt a model to generate conceptual designs of a plaza, park, or street (Guridi et al., 2024; Dubey et al., 2024). Yet, generative models frequently default to learned global priors, potentially reproducing biases or neglecting local cultural markers (Hanna et al., 2024; Jin et al., 2024; Kirk et al., 2024; Sorensen et al., 2024). Our LIVS dataset is explicitly curated to capture local, intersectional preferences in a domain where the geometry, aesthetics, and sociocultural elements of a public space are all crucial (Jacobs, 1961; Gehl & Svarre, 2013; Mitrašinović & Mehta, 2021; Conitzer et al., 2024; Guridi et al., 2024; Dubey et al., 2024). This approach aids in systematically evaluating how T2I alignment can be guided by multiple, sometimes conflicting, user-defined criteria.

## 2.4. Multi-Criteria Preference Learning

Beyond single-objective alignment, multi-criteria preference learning integrates multiple attributes into a unified training signal (Liu et al., 2019; Bhatia et al., 2021; Fan et al., 2023; Chakraborty et al., 2024). This approach has been explored in text-based RLHF, where models are optimized for multiple constraints, such as helpfulness and safety (Kirk et al., 2024; Jin et al., 2024). Recent advancements extend this paradigm to T2I generation by incorporating diverse human preferences.

Prior work has demonstrated the efficacy of multidimensional preference learning in T2I. Zhang et al. (Zhang et al., 2024b) introduced the Multi-dimensional Preference Score (MPS), a model trained on over 918,000 human preference choices across more than 607,000 images, capturing multiple evaluation criteria such as aesthetics, semantic alignment, detail quality, and overall assessment. Similarly, Xu et al. (Xu et al., 2023) proposed *ImageReward*, a reward model trained on 137,000 expert comparisons to encode

human preferences and optimize diffusion models for improved alignment with human expectations. Furthermore, Kuhlmann-Joergensen et al. (Kuhlmann-Joergensen et al., 2025) emphasized the limitations of simplistic preference annotations, advocating for richer human feedback mechanisms to refine T2I model performance and safety.

Building on these developments, we extend multi-criteria preference learning to intersectional urban design goals. We employ the DPO method (Wallace et al., 2023; Rafailov et al., 2024) to fine-tune a T2I model using pairwise preference data that account for accessibility, safety, comfort, invitingness, inclusivity, and diversity. By integrating structured human feedback across multiple criteria, we aim to align generative models with complex societal values in urban planning.

## 3. Methodology: Building the LIVS Dataset

We employed a community-based participatory approach to integrate the local perspectives and experiences that are often absent in top-down, universal datasets (Sieber et al., 2024a; Hosking et al., 2024). This methodology positions community members as collaborators, allowing for the identification of context-specific needs and priorities, such as nuanced understandings of accessibility and safety (Arnstein, 1969; Anttiroiko & de Jong, 2020; Mushkani et al., 2025a). By involving diverse local organizations throughout the process, we aimed to capture intersectional viewpoints and mitigate the risk of imposing external definitions of inclusive design (Costanza-Chock, 2020). This approach also fosters mutual learning, wherein participants gain insights into AI technologies while researchers obtain domain-specific knowledge critical for aligning generative models with real-world public space requirements (Engeström, 2014).

### 3.1. Community Outreach and Engagement

**Initial Contacts.** We contacted 100 community organizations in a mid-sized city (Montreal, population ∼2 million) (Gouvernement du Canada, 2022). These organizations included neighborhood councils, disability-focused nonprofits, faith-based groups, youth advocacy networks, and other local civic stakeholders. Our objective was to obtain a demographically and experientially diverse set of participants who frequently interact with local public spaces (IRCGM, 2018; Creswell & Creswell, 2022).

**Participatory Action Research (PAR).** In line with the principles of PAR, our community-based approach centers on iterative, collaborative inquiry and reciprocal learning throughout the dataset development process (Israel et al., 1998; Cornish et al., 2023). By involving local organizations as active co-researchers, we ensured that the framing of inclusion, safety, and other design criteria emerged from lived

experiences rather than external prescriptions. This iterative feedback loop aligns with PAR's emphasis on collective problem-solving and empowerment, as participants guided each stage of data collection and model evaluation while gaining familiarity with T2I technology and its potential applications in urban contexts.

**Workshops & Interviews.** Over two years, we conducted multiple forms of engagement: eleven workshops, five batches of annotations, and 34 interviews. The process began in 2023 with three multi-stakeholder workshops aimed at collaboratively defining what "equitable design" means for local public spaces. Participants from diverse backgrounds reviewed images of Montreal's public spaces and discussed attributes they considered most important for inclusion. Figure 1 shows the distribution of participants' self-declared demographics.

- *Introductory Sessions (2 workshops, 25–35 participants each):* Provided an introduction to AI and T2I technology. Participants also shared open-ended feedback on the study, AI, and their experiences in local public spaces.

- *Criteria Brainstorming (6 workshops, 28 participants total):* We projected 16 images of existing public spaces in pairwise comparisons and asked participants to describe their reactions. This process generated 634 initial concepts related to inclusivity, accessibility, safety, and other aspects of urban design (see Appendix C for additional details).

- *Validation (1 workshop, 18 participants + 34 interviews):* The 634 concepts were consolidated into 35 intermediate criteria through merging and semantic grouping. Participants then ranked and refined these, producing six final criteria—Accessibility, Safety, Comfort, Invitingness, Inclusivity, and Diversity—based on importance and local relevance.

**Prompting and Early Feedback.**

- *Prompting (1 workshop, 24 participants):* Five groups, each composed of 2-3 citizens, a computer scientist, and an urban architect, collaboratively generated 440 prompts reflecting a variety of public-space scenarios and features. For instance, one group focused on designing prompts for pedestrian promenades with green spaces and safe-street initiatives in historical neighborhoods (see Appendix E for additional details). While refined prompting techniques can shape generative outputs, universal keywords alone may fail to capture local sociocultural and historical contexts (Anttiroiko & de Jong, 2020; Beebeejaun, 2016; Talen, 2012).

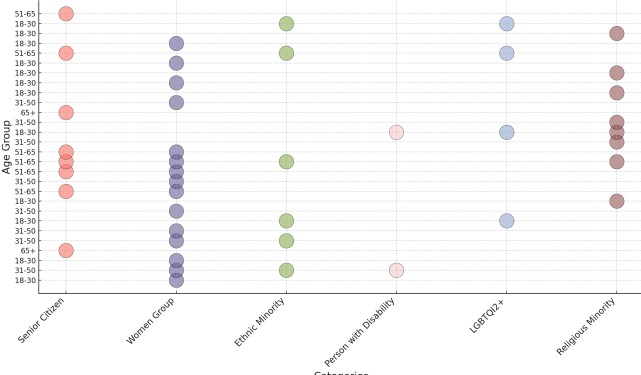

Figure 1. Distribution of participants' self-declared demographics. This figure summarizes the demographic profiles (e.g., age, gender, race/ethnicity, and disability) of the individuals who participated in workshops and annotation activities (Mushkani et al., 2025b).

The LIVS dataset was developed to incorporate granular, community-generated perspectives on accessibility, safety, and inclusivity. By embedding localized knowledge, we aim to reduce the risk of producing uniform design standards that overlook intersectional needs (Low, 2020; Madanipour, 2010; McAndrews et al., 2023).

- *Tutorial and Feedback (1 workshop, 20 participants):* Using prompts from the previous workshop, we generated initial images with Stable Diffusion XL (Podell et al., 2023). Four groups, each comprising 3–4 citizens and an urban architect, tested the annotation platform and provided feedback on visual fidelity and representation before the larger-scale annotation phase (see Appendix C.3, Figure 12 for annotation platform). To optimize data quality and minimize cognitive fatigue, we randomly presented three of the six criteria in each comparison. During pilot trials, participants found evaluating all six criteria difficult, which reduced their sense of progress and visual engagement. The multiple rating elements also constrained image size, making it harder to assess spatial details. Focusing on three criteria enabled more meaningful engagement. Over successive annotation batches, we ensured that all six criteria were robustly evaluated.

**Annotations and Evaluation.**

- *Annotations (5 batches, 18 participants):* The annotation tasks were divided into five batches, each lasting two weeks and containing approximately 750 pairwise comparisons per participant, totaling 42,235 raw comparisons. In each task, two images were displayed side by side with three randomly selected criteria from the six. Annotators used a slider ranging from $-1$ (strong preference for the left image) to $+1$ (strong

preference for the right image), with 0 indicating neutrality. An open-source annotation platform was developed to facilitate this process, featuring a user-friendly slider interface to accommodate diverse backgrounds (see Appendix D for additional details). A multi-stage data-cleaning process refined the dataset, including removal of quality-control annotations and incomplete submissions. The resulting dataset comprises 35,510 high-quality annotations.

- *Evaluation (1 workshop):* After the annotation phase, we fine-tuned Stable Diffusion XL using these data. Participants then evaluated the fine-tuned model's outputs, discussing alignment with local norms and values.

## 3.2. Criteria Consolidation

The original 634 concepts spanned physical, social, and psychological attributes of public spaces (e.g., lighting, presence of diverse user groups, multilingual signage, seating). Through collaborative merging, collective voting, and iterative discussion, participants identified six *core criteria* (Figure 2 illustrates this process):

**Accessibility:** Physical and cognitive usability for people of all abilities, including elevators, ramps, tactile indicators, and clear signage.

**Safety:** Freedom from crime, hazards, or harassment, often reflected in well-lit areas, clear visibility, active presence, or protective barriers.

**Comfort:** Availability of amenities (e.g., seating, shade) and mitigation of environmental conditions (e.g., noise, temperature), including cleanliness and spatial shelter.

**Invitingness:** Features that encourage people to enter and remain, such as greenery, open layouts, welcoming signage, or visible communal areas.

**Inclusivity:** Avoidance of exclusionary design; support for cultural or religious needs; signage in multiple languages.

**Diversity:** Representation of different demographic groups and a range of potential uses.

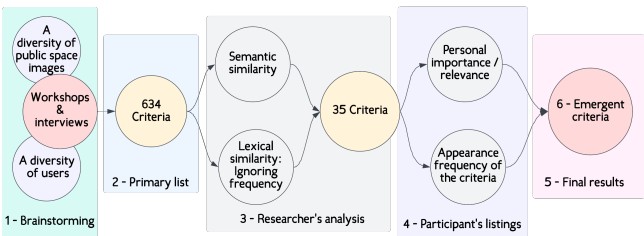

*Figure 2.* Distilling the initial concepts into six core criteria. The figure shows how 634 distinct ideas were iteratively merged, discussed, and ranked to arrive at final high-level categories.

Participants viewed these six criteria as sufficiently comprehensive to represent inclusive urban design (Anttiroiko & de Jong, 2020; Mitrašinović & Mehta, 2021), while emphasizing that intersectional dimensions (e.g., disability + race/ethnicity) were woven throughout each category.

## 3.3. Prompt and Image Preparation

**Prompt Collection.** Initially, a total of 440 textual prompts describing local public-space scenarios were collected from the *Prompting* workshop. Figure 3 presents a word cloud that illustrates the distribution of keywords within these prompts. To expand the dataset and ensure a sufficient variety of prompts, we utilized workshop transcripts and employed a large language model (GPT-4o (OpenAI et al., 2024)) with three distinct prompting strategies (see Appendix C.1 for additional details). This process generated approximately 2,910 synthetic prompts, encompassing diverse public-space typologies, amenities, and contexts. We assessed the differences between human-written and LLM-generated prompts using Jensen-Shannon Divergence (Plank & van Noord, 2011), which indicated that all three strategies produced diverse outputs and contributed to a comprehensive range of scenarios.

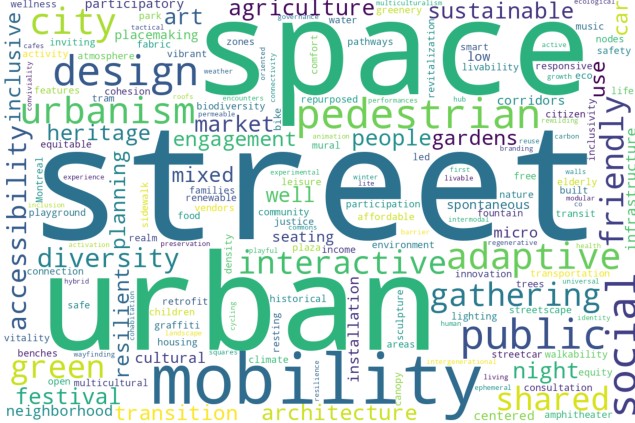

*Figure 3.* Word cloud depicting the frequency of various concepts within the 440 collected prompts. The size of each word reflects its prevalence, highlighting key themes such as *public-space typologies*, *amenities*, and *contextual use scenarios*. This distribution underscores the diversity and contextual comprehensiveness of the prompt dataset.

**Image Generation.** To maximize image diversity and avoid comparisons between images that are too similar, we generated for each prompt up to 20 images using Stable Diffusion XL (Podell et al., 2023). We then selected the four most distinct images per prompt using a greedy algorithm based on CLIP similarity scores (see Appendix 1 for additional details). In total, 16,693 images were generated, with a subset set aside for quality checks (see Appendix D for additional details), leaving 13,462 images for annotation.

For each image pair, participants were randomly assigned three of the six criteria and required to annotate at least one criterion. This modification, implemented following

feedback from the evaluation workshop, ensured that each criterion had an equal opportunity to be evaluated. However, participants reported that certain criteria, such as *Inclusivity*, were more subjective, leading to fewer distinct preferences and a higher incidence of neutral or equal annotations. Consequently, while each criterion had an equal chance of being selected, some received fewer definitive annotations compared to others. This is reflected in Figure 4, where *Inclusivity* received the fewest annotations, whereas *Comfort* and *Invitingness* were annotated more frequently. Overall, the annotation process yielded 37,710 image-level comparisons, amounting to approximately 113,130 criterion-level annotations. Though modest in size by machine learning standards, the dataset prioritizes quality over volume.

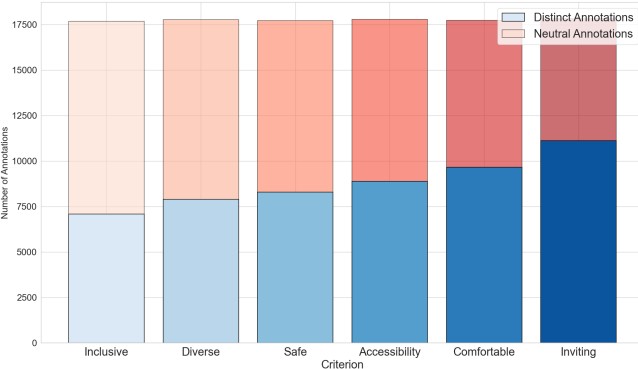

*Figure 4.* Distribution of annotation frequencies for each criterion after data cleaning. Red indicates neutral or equal preferences, while blue represents distinct preferences.

### 3.4. Ethical Considerations

All activities were approved by a research ethics board and followed a co-production methodology with community organizations (IRCGM, 2018). As part of a knowledge exchange and capacity-building process, we learned about public space use and perception from participants, while they gained insights into AI and its applications in urban contexts. Participants were compensated for their involvement and provided written informed consent for data collection. Personally identifiable information was anonymized to ensure confidentiality (Creswell & Creswell, 2022). During the annotation sessions, some AI-generated images exhibited biases or inaccuracies (Hosking et al., 2024; Turchin, 2019b), prompting revisions to the data-collection protocol. To promote equitable participation, we developed an accessible web interface that accommodated annotators from diverse backgrounds, including those with disabilities, ensuring effective evaluation of images.

## 4. Experiments with LIVS

We present four case studies examining how multi-criteria feedback from LIVS can guide text-to-image (T2I) align-

ment. Detailed implementation notes (including hyperparameters, prompt generation, and data processing) appear in the Appendix F.

### 4.1. Case Study I: Does Multi-Criteria DPO Improve Alignment?

**Setup.** We fine-tuned Stable Diffusion XL (SDXL; Podell et al. 2023) using Direct Preference Optimization (DPO; Rafailov et al. 2024). Each pairwise comparison in LIVS was treated as a binary "preferred" vs. "not preferred" signal. When annotators provided split ratings across criteria (e.g., preferring the left image for *Safety* but the right image for *Inclusivity*), we collapsed the feedback via majority voting. This procedure trained a single model on all criteria and evaluated its performance on each criterion individually. Model outputs were then generated for a held-out set of prompts and compared against the SDXL baseline.

**Results.** Out of 2,200 new comparisons, annotators chose the DPO-aligned model in 700 (32%) instances and the baseline in 300 (14%), marking the remaining 1,100 (50%) as neutral (See Appendix I). Criteria with more training annotations (e.g., *Comfort*, *Invitingness*) showed stronger improvements under DPO, whereas *Inclusivity* and *Diversity* had higher neutral ratings (Figure 6). Qualitative inspection revealed that DPO produced clearer walkways and seating configurations but did not consistently generate detailed features (e.g., ramps, tactile paving, multilingual signage). Examples of baseline, single-criterion, and multi-criteria fine-tuned outputs are shown in Figure 7. Overall, the results suggest that DPO improves the alignment of T2I outputs with local preferences but continues to exhibit substantial variability across criteria. The high proportion of neutral ratings indicates that DPO alone is insufficient to capture the complexity of community preferences, underscoring the need for more nuanced and adaptive alignment algorithms (see Appendix G for additional details).

**Additional Observations.** Comparing Figure 4 (overall dataset) with Figure 5 (evaluation set) suggests a similar distribution of annotations and neutral responses: criteria with more training annotations (e.g., *Comfort*, *Invitingness*) generally exhibit stronger DPO preference in the evaluation. This pattern indicates that additional data may further improve multi-criteria alignment through DPO. However, we currently do not leverage neutral labels, and majority voting in multi-criterion comparisons may overlook nuanced disagreements across criteria. Specifically, when annotators provide conflicting preferences across the three assigned criteria, the final label is determined by majority vote—e.g., two preferences for the left image and one for the right results in a binary label favoring the left. This simplification disregards the internal variation across criteria and may

obscure which dimensions drive preference. Future work could explore methods for incorporating neutral ratings and resolving multi-label conflicts without collapsing them into binary outcomes. Further analysis of criterion-level variance is provided in Appendix I.

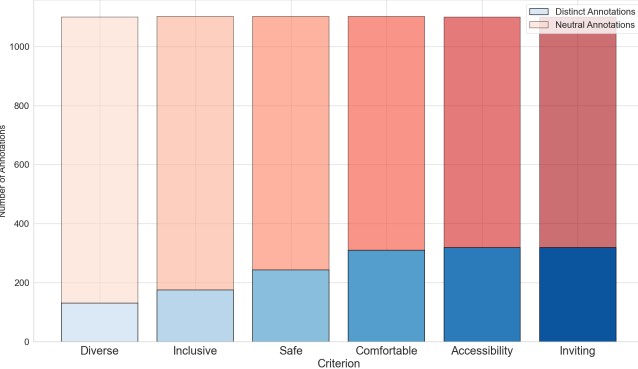

*Figure 5.* Criteria distribution on the new evaluation dataset. Neutral ratings were more common for *Inclusivity* and *Diversity*, indicating subtler or more subjective distinctions in these categories.

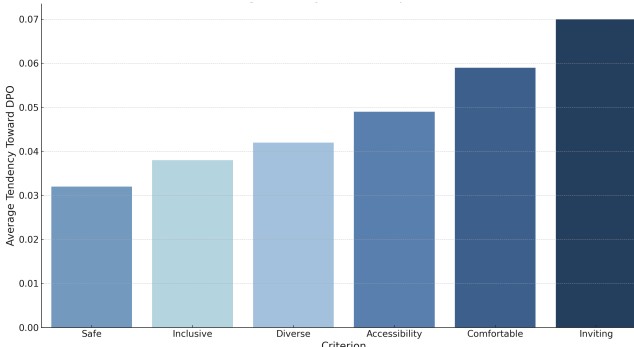

*Figure 6.* Average DPO preference by criterion. Bars show the mean tendency toward DPO-aligned outputs. Performance tends to improve in criteria with higher annotation counts.

## 4.2. Case Study II: Do Preferences Vary Across Identities?

**Setup.** We examined whether participant demographics (e.g., disability status, age) influenced choices between DPO-finetuned and baseline outputs. We aggregated each individual's total count of DPO-favoring versus baseline-favoring comparisons across the six LIVS criteria.

**Results.** Most participants showed a modest preference for DPO, although two late-joining individuals rated the baseline and DPO models similarly (Figure 8). These participants joined the study after the core workshops and did not participate in earlier collaborative sessions that established the six final criteria. Their equal preference may indicate that less involvement in the knowledge-exchange process can lead to different or less pronounced alignment percep-

Prompt: A quiet, inclusive meditation garden in a busy urban area.

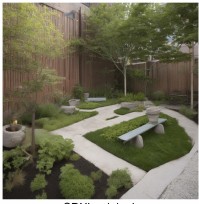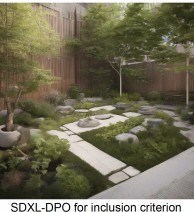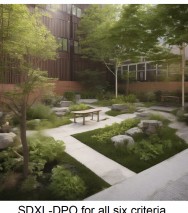

SDXL original     SDXL-DPO for inclusion criterion     SDXL-DPO for all six criteria

*Figure 7.* Three variants of SDXL outputs generated with the same prompt, seed, and hyperparameters. **Left:** Baseline SDXL lacks cohesive pathways and shows minimal accessibility features. **Middle:** An inclusivity-finetuned model adds partial signage and barriers but still has uneven paths. **Right:** A multi-criteria finetuned model shows smoother transitions, clearer walkways, and additional seating.

tions. Some annotators who reported mobility challenges were more likely to favor the DPO outputs, suggesting that DPO captured partial accessibility-related cues. However, no single demographic factor dominated preferences. This variation underscores the importance of collecting intersectional data rather than applying a universal alignment rule.

## 4.3. Case Study III: Does Prompt Composition Affect Rating Consistency?

**Setup.** We compared images generated from 440 human-authored prompts with four GPT-4o-generated prompt sets. Both baseline SDXL and the DPO-finetuned model were used for each prompt. Annotators then compared image pairs on a random subset of three criteria.

**Results.** Human-authored prompts (Method 0) led to fewer neutral ratings, suggesting they produced more visually distinct outcomes (Figure 9). In contrast, GPT-4o-generated prompts (Methods 1–4) exhibited higher rates of neutrality, possibly due to reduced contextual specificity. As shown in Figure 9, the proportions of neutral annotations vary across methods, indicating that prompt design significantly influences annotators' perceptions of alignment differences.

## 4.4. Case Study IV: Do Intersectional Identities Rate SDXL Outputs Differently?

**Setup.** We explored whether different intersectional identities (e.g., disability status × race/ethnicity) assigned systematically distinct raw scores to images across LIVS criteria. Each participant rated images for multiple criteria in separate annotation tasks.

**Results.** We observed that individuals from various intersectional groups (e.g., participants with mobility challenges) typically assigned lower *Accessibility* or *Safety* scores. This variation underscores the need for local, intersectional feed-

back in T2I alignment for urban planning, as a single global objective cannot capture such diverse preferences. See Appendix Section H.1, Figures 16 and 17 for further details.

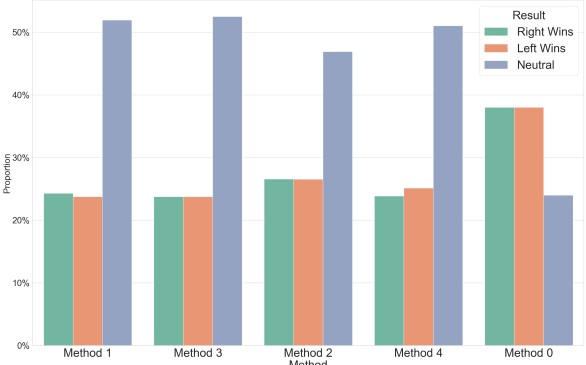

Figure 9. Proportion of neutral annotations for human-written (Methods 0) versus GPT-4o-generated prompts (Methods 1-4). Method 0 indicates that human-authored prompts elicited more decisive preferences (fewer neutral annotations), suggesting clearer visual distinctions

interpreting neutral responses as alignment failures, we consider them indicative of alignment adjustments that were either too subtle or inadequate to capture the complexity of participant feedback. Prompt variations also contributed to this ambiguity, as some prompts yielded less distinct visual differences. Overall, neutral ratings emphasize the importance of richer, context-sensitive alignment methods that consider both prompt content and multi-criteria data, including conflicting user needs, and highlight the need for further research (Hosking et al., 2024; Sorensen et al., 2024).

**Methodological Constraints and Alignment Strategy.** Because DPO requires binary labels, we collapsed multi-criteria feedback into single preference signals during training. This simplification overlooks intra-sample disagreement and limits the model's capacity to capture the full complexity of pluralistic preferences. Future alignment strategies should accommodate multi-dimensional annotations directly, enabling more nuanced modeling of diverse community feedback.

**Intersectionality and Local Variation.** The variations in preference highlight the potential shortcoming of single-objective alignment. Pluralistic alignment attempts to unify multiple local perspectives, but fully reconciling them may be infeasible in a single model. One potential future direction is the development of *user-personalized alignment layers*, where a single base T2I model can adapt to specific subgroups or contexts (Jang et al., 2023; Kirk et al., 2024; Sorensen et al., 2024).

**Comparison with Overton and Distributional Pluralism.** Sorensen et al. (Sorensen et al., 2024) propose Overton pluralism, steerable alignment, and distributional pluralism to address heterogeneous human values. Our multi-criteria, participatory framework echoes Overton pluralism by capturing a broad range of locally valid design norms, while intersectional feedback supports distributional coverage for

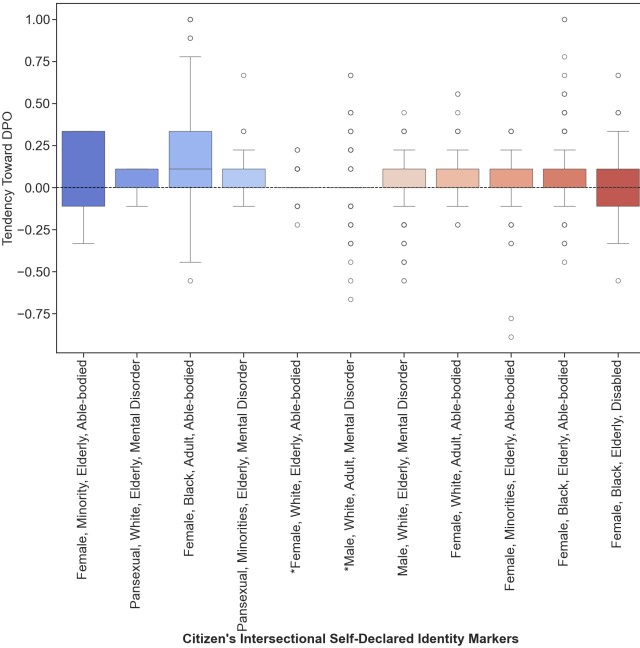

Figure 8. Boxplot of participant preferences. A score of 1 indicates a tendency towards DPO, while a score of -1 indicates a tendency towards the base model. Two late-joining participants (asterisks) showed no clear preference, whereas most favored DPO to some extent. One participant with a disability indicated a smaller margin of preference for DPO, reflecting individual-level variability.

## 5. Implications

Our findings inform pluralism and multi-objective machine learning. Implications follow:

**Multi-Criteria Overlaps and Conflicts.** Some participants valued certain criteria (like Diversity) above others (such as Comfort). Others perceived Safety and Comfort as interlinked, finding it hard to separate physical hazards from environmental conditions. Future alignment methods might benefit from hierarchical, weighted, context-sensitive, or dynamic objective formulations, capturing partial dependencies among criteria (Gabriel & Ghazavi, 2021; Stray, 2020; Li et al., 2024; Sorensen et al., 2024).

**Neutral Annotations as a Signal.** Approximately half of the final evaluations were rated as neutral. While these outcomes may initially suggest a lack of meaningful improvement, they may also reflect genuinely balanced preferences or participants' difficulty in discerning visual cues for certain criteria—particularly Inclusivity and Diversity. In several interviews, participants noted that subtle or symbolic elements were not always visually salient. Rather than

varied subgroups. Unlike text-only methods, visuals can convey subtle spatial details (e.g., ramps or diverse crowds) that reduce ambiguity and highlight group-specific preferences. By combining these signals with DPO, we also advance Overton's commitment to retaining multiple permissible viewpoints, although roughly half of our comparisons remained neutral—reflecting persistent tensions and underscoring that no single, static alignment fully resolves all local conflicts.

**Broader Relevance.** While our application domain is urban planning, the core methodology—community-centered, multi-criteria preference data, and DPO-based fine-tuning—applies broadly to scenarios where local norms matter (e.g., cultural heritage preservation, healthcare, educational content creation) (Kirk et al., 2024; Huang et al., 2024a; Hosking et al., 2024; Harland et al., 2024).

# 6. Conclusion

We presented *LIVS*, a dataset and methodology for pluralistic alignment of T2I models centered on intersectional, local feedback for inclusive public spaces. Our experiments with DPO fine-tuning on Stable Diffusion XL revealed moderate preference gains relative to a base model, especially under criteria such as Invitingness or Accessibility. Nonetheless, roughly half of the annotations were neutral, underscoring the complexity of reconciling diverse criteria and identities in a single generative alignment objective.

**Contributions.** We introduced a multi-year, participatory process that established six locally validated criteria for inclusive urban design, demonstrated how DPO can incorporate multi-criteria signals, and provided empirical evidence of partial alignment success. Additionally, we showcased how intersectional feedback can highlight local or demographic nuances that global alignment schemes may overlook. The LIVS dataset and model enable two key use cases: first, supporting empirical research on what constitutes inclusivity in urban design by providing structured annotations on comfort, accessibility, and other public space attributes; and second, facilitating democratic deliberation in public space renovation projects through visualizations that allow policymakers and communities to explore context-specific interventions. These use cases illustrate how generative models aligned with LIVS can bridge technical capabilities with situated local knowledge, fostering equitable and participatory urban design. See Appendix B for more details.

**Limitations.** Our dataset concentrates on one mid-sized, multicultural city, which is well-suited for pluralistic alignment but restricts the range of local norms represented. Other contexts may have profoundly different needs or design principles. Additionally, the total number of participants was relatively low, and our final test set of 2,200

annotations remains modest compared to the complexity of T2I generation. Although our results show promising alignment gains, scalability to larger, more diverse regions or to other policy domains is not guaranteed.

**Future Work.** Future research might explore multi-objective optimization by extending beyond pairwise DPO to address correlated or competing criteria, such as Comfort and Safety, potentially using methods like Pareto optimality or weighted objectives (Chakraborty et al., 2024). Investigating neutral annotations could involve developing refined strategies, such as partial reward signals, to interpret neutral feedback more effectively instead of discarding it as non-informative. Additionally, developing tools for policy integration in collaboration with urban planning agencies may facilitate the application of T2I alignment in real-world policy decisions through interactive tools for stakeholders. Enhancing personalization by experimenting with adaptive alignment tailored to specific subgroups or contexts, especially where intersectional dimensions are important, might improve model relevance and inclusivity. Overall, our findings suggest that a pluralistic alignment paradigm could be promising for creating locally grounded, inclusive generative AI systems by acknowledging intersectional viewpoints and supporting multi-criteria feedback loops to better serve diverse community needs.

**Availability.** The LIVS dataset—including citizen-provided self-identification markers (with consent)—is available for research purposes at mid-space.one. This release aims to establish a benchmark for pluralistic alignment in text-to-image generation and supports both criterion-specific and user-specific customization. Future work can leverage these granular annotations to develop personalized models and adaptive fine-tuning strategies that more effectively address the unique needs of diverse user groups.

# Impact Statement

This study adapts T2I alignment to an urban context, aiming to better represent diverse demographic perspectives. The dataset and results may facilitate more inclusive public-space design. However, there is a risk of oversimplifying complex social challenges by attempting to visualize inclusivity and diversity, which are multifaceted. Moreover, potential biases in the annotated data could affect how the model prioritizes certain user groups. We view our approach as one step toward more fine-grained, democratically informed generative AI, while recognizing that deeper societal involvement and critical oversight remain essential.

# Acknowledgements

We thank Emmanuel Beaudry-Marchand for developing the foundational schema of the Aipithet Platform used during

the annotation process. We also acknowledge Toumadher Ammar for her assistance with the workshops, and Jerome Solis for his support with coordination. We are grateful to Sarah Tannir, Leandry Jieutsa, Adèle Kremer, Frédérique Roy, and Roxane Kasprzyk for their contributions.

We appreciate the collaboration of our partner organizations and architectural offices, including Enclume, Sid Lee Architecture, Dark Matter Labs, IVADO, and the Canadian Commission for UNESCO.

This research was funded by the *Soutien aux initiatives avec les collectivités et les entreprises – Collaboration avec les organismes communautaires* program of the *Université de Montréal*, and the *Bourse IA ESP*. Additional support was provided by *Mitacs* and the *Fonds de recherche du Québec – Société et culture (FRQSC)* Doctoral Scholarship.

We thank all 30 community organizations in Montreal that supported citizen collaboration. The following were among the most engaged and actively involved: the Congolese Community Center of Montreal, Altergo, La Maisonnée, Cummings Centre, Projet Changement, Women's Center of Plateau Mont-Royal, LGBTQ+ Community Center of Montreal, Marguerite-Bourgeoys Hub, RÉZO, Afrique au Féminin, L'Agence On est là!, and the Montreal Women's Groups Table.

This work was enabled in part by computing resources provided by Mila.

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

# A. LIVS Dataset Viewer

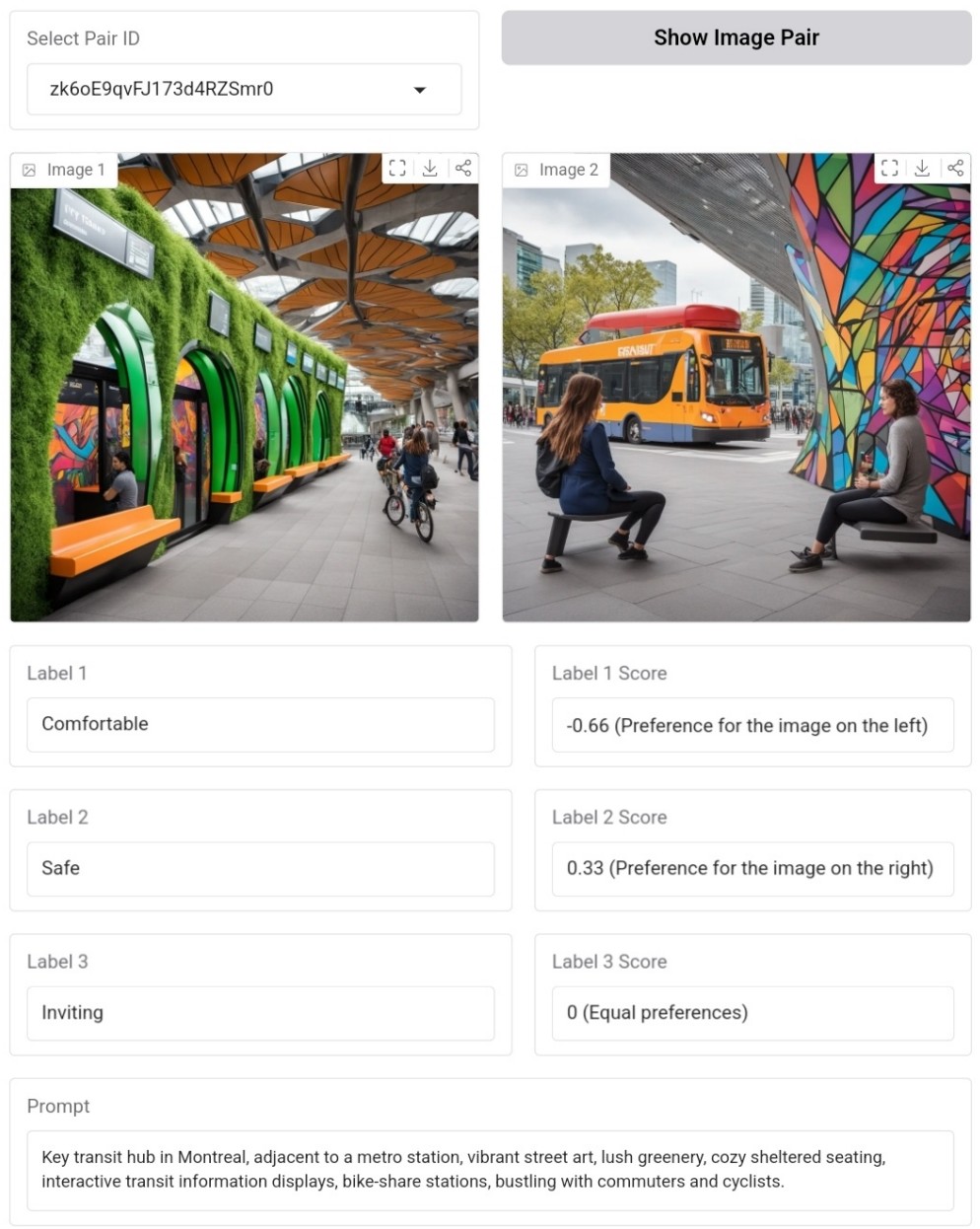

*Figure 10.* An overview of the dataset, illustrating pairs of images for each prompt alongside corresponding preference scores.

# B. Use Cases

The multi-criteria alignment approach described in the main paper can be adapted for various real-world scenarios in urban planning and beyond. Below, we outline specific applications in which intersectional alignment and rapid text-to-image generation may offer practical benefits.

**Community Consultations.**    Local governments or urban planners often conduct participatory design sessions for proposed public spaces. The aligned model can generate visual scenarios that reflect multiple local criteria, such as accessibility or safety. Community members can then provide feedback on which visualizations best meet their needs, potentially reducing barriers for stakeholders unfamiliar with technical planning diagrams.

**Peer-to-Peer Discussion.**    Community members can use the model to explore differing priorities in contested designs. By quickly producing multiple variations of a layout, participants can discuss trade-offs (e.g., balancing comfort with affordability) without relying on professional mediators.

**Rapid Visualization of Ideas.**    Urban designers and architects can employ the model to iterate on design concepts at an early stage, generating a variety of sketches that incorporate multi-criteria feedback (e.g., inclusivity or invitingness). This process can reveal overlooked aspects before substantial resources are committed.

**Teaching and Education.**    In architecture or urban-planning courses, students can interact with the model to see how different prompts and annotation signals affect output images. This hands-on experience can clarify alignment methods and potential biases in generative models.

**Amplifying Marginalized Voices.**    Historically excluded groups (e.g., people with disabilities, marginalized ethnic communities) can utilize the model to communicate spatial requirements, such as multilingual signage or wheelchair accessibility. Visual prototypes allow direct articulation of needs and can lead to more inclusive design outcomes.

# C. Further Details on Methodology: Building the LIVS Dataset

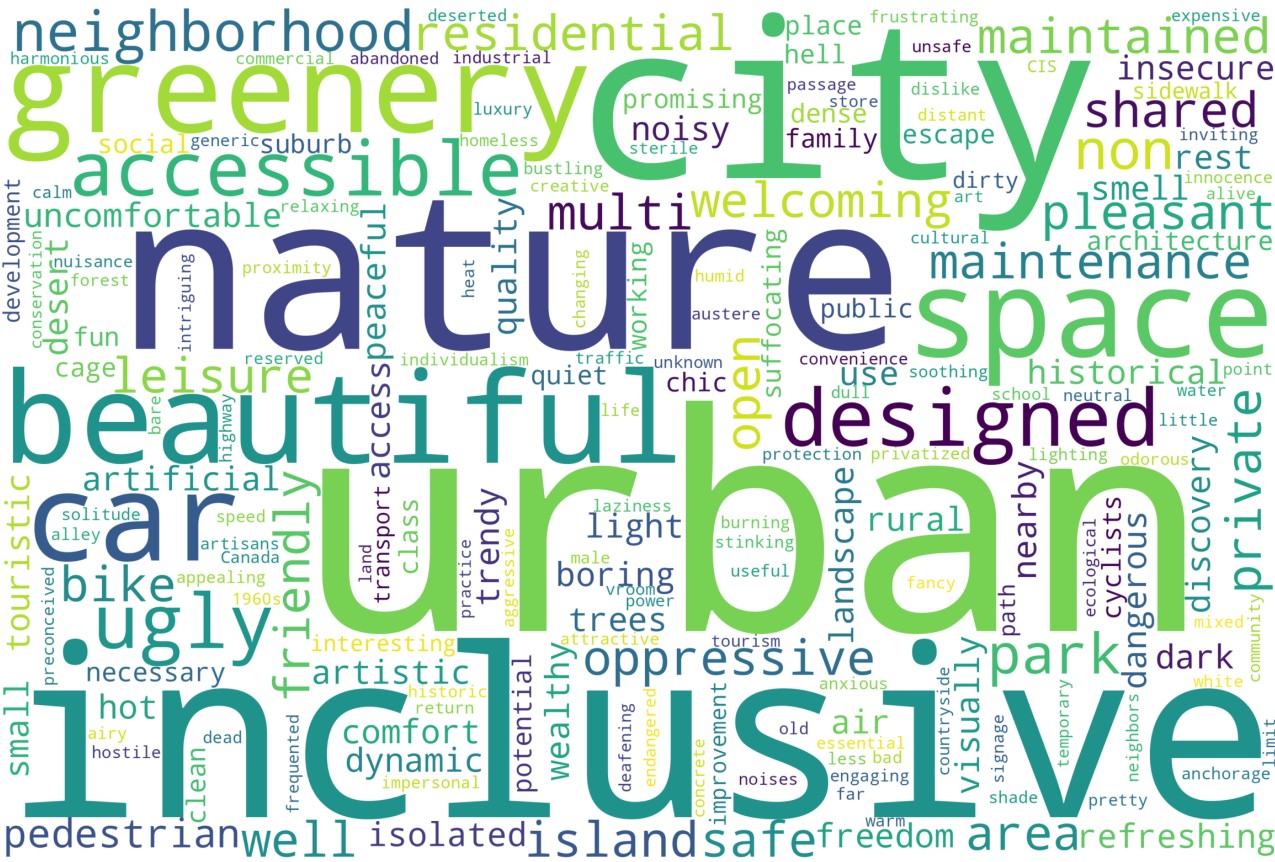

*Figure 11.* Word cloud of the 634 collected public-space attributes that were consolidated into six high-level criteria. Larger words appear more frequently among the original attributes.

## C.1. Prompt Augmentation

We collected 440 prompts from annotators, each representing distinct scenarios related to public space typologies, amenities, and ambiances in Montreal. To expand the dataset, we supplemented these with synthetic captions generated by GPT-4o (OpenAI et al., 2024). To ensure diversity, we incorporated a wide range of concepts drawn from public space literature, such as typologies like parks and wide walkways, amenities such as seating areas and streetlights, natural elements like forests and vegetation, locations such as suburban Montreal and old industrial ports, people such as First Nations and children, architectural elements like duplexes and houses, transportation modes including bike lanes and trams, artistic features such as street murals and art sculptures, different times like winter, nighttime, and Christmas, and animals such as dogs and raccoons. These concepts were used to increase the diversity of the concepts in the generated prompts. We employed three distinct prompting strategies to ensure that the synthetic prompts retained similarity to those generated by humans while covering a wide range of topics relevant to public space design. The specific strategies and prompts used for the LLM are outlined below.

**Method 1** We randomly sampled 8-16 prompts from the human-generated set and used them as in-context examples for the LLM to generate new prompts.

**System prompt used Method 1**

Your task is to craft detailed and imaginative prompts suitable for diffusion models like Stable Diffusion. These prompts should generate images illustrating the variety of Montreal's public spaces, capturing the community's diverse aspirations and values.
Each prompt must be rooted in a specific scenario related to Montreal's public spaces. You will be provided with the scenario, keywords, and examples of prompts related to these scenarios. Using this information, your task is to create a series of diverse, con¯textually rich, and relevant prompts following a style similar to the ones given as examples. These should aim to generate images showcasing Montreal's public spaces from varied perspectives.

**Method 2** We provided the LLM with a detailed scenario which was also provided to the annotators during the initial prompt collection phase. Along with this we also provided several keywords related to the public space concepts mentioned earlier. Additionally, we included 8 randomly selected in-context samples relevant to the scenario guiding the model to generate new prompts based on these concepts.

**System prompt used Method 2**

Your task is to craft detailed and imaginative prompts suitable for diffusion models like Stable Diffusion. These prompts should generate images illustrating the variety of Montreal's public spaces, capturing the community's diverse aspirations and values. To achieve this, you will construct prompts using specific keywords provided for the following categories:
Typology: The type of spaces you want to depict
Elements: Distinct elements to include in your scene
Context: The scenarios in which your elements are placed
Style: The artistic style or technique that the image should emulate, defining its visual appearance
Mood: The overall mood or atmosphere of the image

You will also be given a few examples that have been generated using these keywords. Using all this information create a complete, coherent prompt similar in style to the examples. Aim for creativity and diversity in your prompts, ensuring they cover several aspects of the keywords given. These should aim to generate images showcasing Montreal's public spaces from varied perspectives.

Note:
1. Ensure your prompts integrate some of the provided keywords to encapsulate the community's desired visions of Montreal's public spaces but ensure that style and length is same as the examples.
2. Do not mention the style and mood explicitly. Use keywords that bring out these attributes naturally.
3. The style and the length of the prompts should be similar to the examples given. The prompt should be less than 77 tokens.

**Method 3** This method used a template-based approach where specific keywords related to public space concepts in the original prompts were masked. We instructed the LLM to replace these masked keywords with concepts from a wide variety of public space themes. This ensured the prompts closely followed the structure of the human-generated ones while incorporating a diverse range of concepts.

---

**System prompt and incontext sample used Method 3**

Your task is to craft detailed and imaginative prompts suitable for diffusion models like Stable Diffusion. These prompts should generate images illustrating the variety of Montreal's public spaces, capturing the community's diverse aspirations and values.

For this task, you will be provided with a templated sentence containing several placeholders. Each placeholder represents a specific category (e.g., [Typology], [Location], [Activity], [Amenity]). Alongside the templated sentence, you will receive a list of words or phrases corresponding to each category. Your objective is to select the most appropriate word or phrase from each list to fill in the placeholders, creating a meaningful and grammatically correct sentence.

The structure of the templated sentence might require minimal modifications to ensure grammatical correctness and cohesiveness once the placeholders are filled.

Example 1:

Template: a [Typology] for [People] in [Location]

Keywords:
Typology: artistic eco friendly park, pedestrian street, all identities, two-story residential street, park, neighbourhood public space, urban square, wide walkway
People: elderly person, adults, first nations, children, teenagers, adults and elderly people, black and white families, a mother and her child, people, various ethnicities
Location: plateau, wellington neighbourhood, old port, montreal 's chinatown, old montreal, Montreal, downtown montreal, mont royal street

Output: A neighbourhood public space for children, teenagers, adults and elderly people in Montreal

¡more examples¿

---

To measure deviation from human-written prompts, we computed the Jensen-Shannon Divergence (JSD) (Plank & van Noord, 2011). Methods 1 and 2 produced slightly higher JSD scores (0.53 and 0.58) than Method 3 (0.40), indicating that all three approaches contributed diverse scenarios, with Method 3 aligning more closely with human style.

## C.2. Image Generation

We used Stable Diffusion XL to generate 20 images per prompt, varying hyperparameters (seed, guidance scale, steps). Because initial user feedback indicated difficulty in differentiating images, we applied a greedy selection strategy to choose the 4 most distinct images based on CLIP similarity scores. Algorithm 1 details this procedure.

## C.3. Annotation Details

Figure 12 shows the web-based annotation interface. Users rated each image pair by moving a slider to the left or right, indicating their preference strength or neutrality. They could rate up to three randomly assigned criteria per comparison, consulting embedded definitions as needed.

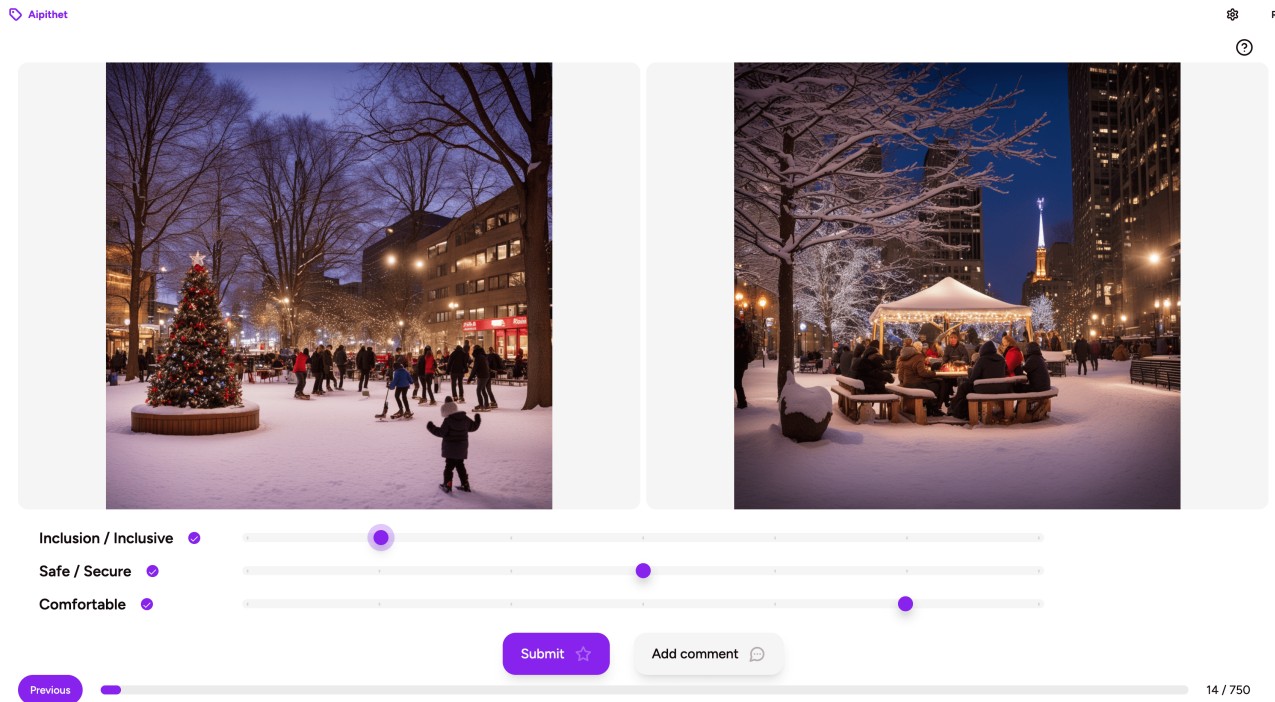

*Figure 12.* Annotation interface for LIVS. Users compared two images on a slider, optionally clicking on the purple dot next to each criterion for its definition.

---

**Algorithm 1** Selecting the 4 Most Diverse Images Using CLIP Similarity Scores

---

1: **Input:** Similarity matrix $S$ (size $n \times n$), number of images $k = 4$
2: **Output:** Indices of the $k$ selected images, $selected\_indices$
3: $n \leftarrow \text{len}(S)$
4: $selected\_indices \leftarrow []$
5: $first\_index \leftarrow \arg\min\big(\text{mean}(S, \text{axis} = 1)\big)$
6: Append $first\_index$ to $selected\_indices$
7: **for** $j = 1$ to $(k - 1)$ **do**
8:     $min\_similarity \leftarrow \infty$
9:     $next\_index \leftarrow -1$
10:     **for** $i = 0$ to $(n - 1)$ **do**
11:       **if** $i \notin selected\_indices$ **then**
12:         $current\_similarity \leftarrow \max(S[selected\_indices, i])$
13:         **if** $current\_similarity < min\_similarity$ **then**
14:           $min\_similarity \leftarrow current\_similarity$
15:           $next\_index \leftarrow i$
16:         **end if**
17:       **end if**
18:     **end for**
19:     Append $next\_index$ to $selected\_indices$
20: **end for**
21: **return** $selected\_indices$

---

# D. LIVS - Annotation Protocol

Your help is crucial for creating an AI model that understands what makes Montreal's public spaces good for everyone. This guide will assist you in comprehending the labeling process, how to conduct labeling, and what to look for during this exercise.

## D.1. Before You Start

- **Read the criteria definitions:** Understand what each criterion (like safety or comfort) signifies before you begin. (See the definitions below.)

- **Take breaks:** Avoid attempting to do too many annotations at once. It's preferable to return refreshed.

- **Take your time:** Ensure the annotations are of high quality. On average, each comparison should take between 15 to 30 seconds.

- **Use a computer:** This task is more manageable on a computer than on a phone or tablet.

## D.2. The Labeling Process

- **Evaluating image pairs:** You will assess each pair of images based on three specific criteria displayed on the webpage. For each criterion, adjust the slider to indicate if the image on the right or left better aligns with that criterion. If neither image fits or both are equally suitable, you may position the slider in the center. However, be aware that center positions provide no distinct preference data.

- **Personal perspective:** Annotate based on your own judgment, experience, and perspective. We value your individual insight and are not seeking an objective assessment.

- **Focus on urban space characteristics:** Your decision should be based solely on the characteristics of the urban space rather than the presence of people or animals.

- **Handling distorted images:** Do not spend additional time making sense of disfigured or unclear images, as these are common with AI-produced imagery.

- **Utilize the commenting feature:** This allows you to add nuances or context to your annotations. Remember, this is voluntary and does not count towards the total number of annotations.

- **Asking questions:** If you're ever unsure about anything, please send us an email at hugo.berard@umontreal.ca.

## D.3. Labeling Duration

You can label the images on a web platform accessible from any location. A code will be sent to your email for access. Simply create a profile using your email and a chosen password. Below is the schedule for image annotation:

- **Duration:** The labeling spans 8 weeks, organized into four 2-week batches.

- **Task:** Each batch requires the annotation of 750 images.

- **Start date:** The labeling begins on 01 May 2024.

- **End date:** Please try to finish 750 annotations before the deadline for each batch. The deadlines are indicated below.

Please note that the platform limits users to 90 annotations per session, with each session lasting about 25 minutes. Completing all annotations for each batch is estimated to take around 4 hours.

**D.4. Labeling Timeline**

- **First batch:** 01 April – 14 May

- **Second batch:** 15 May – 28 May

- **Third batch:** 29 May – 11 June

- **Fourth batch:** 12 June – 20 June

**D.5. Definitions**

- **Public space:** Public space is an area where everyone can go, like parks and streets, designed for people to meet, play, and relax together in cities and towns.

- **Inclusion (Inclusive):** Spaces where everyone is welcome and feels respected. These are places that do not discriminate against anyone.

- **Safe / Secure:** Spaces where public safety is ensured through various measures. Spaces where one feels calm and safe, free from dangers related to physical elements, pollution, or any other concerns that could diminish a sense of security.

- **Comfortable:** Well-equipped spaces with quality facilities that provide material comfort; places where one feels at ease and protected from the elements.

- **Inviting:** Spaces that attract and engage people through appealing elements and activities; places that encourage community participation and interaction.

- **Diverse:** Spaces that cater to the diversity of social groups and to the variety of services, activities, and functions. These are places offering a range of uses and meeting the needs of different cultures, ages, and abilities.

- **Accessibility:** Urban spaces that are easily accessible and navigable for everyone, regardless of physical ability. These include features such as ramps, wide walkways, clear signage, and tactile indicators for safe and convenient access throughout the area.

# E. Prompting Workshop Details

**Questions for the Prompt Creation**

- What are the surroundings?

- What decorative features and objects does the place have?

- Is there nature present, and if yes, what kind?

- How are the weather and light conditions?

- What is the composition of the image?

- What materials and surfaces are present?

- How would you describe the atmosphere?

- What amenities should be present in the space?

**Questions for the Evaluation of the Images**

- Can you imagine using the public space yourself?

- Does the public space match what you had imagined when creating the prompt?

- Are you satisfied with the image?

- Can you see the image being used as a design for a public space in real life?

**Groups**

**Group 1**

- First hands-on session – Scenario A
- Second hands-on session – Scenario B
- Optional – Scenario F

**Group 2**

- First hands-on session – Scenario B
- Second hands-on session – Scenario C
- Optional – Scenario G

**Group 3**

- First hands-on session – Scenario C
- Second hands-on session – Scenario D
- Optional – Scenario H

**Group 4**

- First hands-on session – Scenario D
- Second hands-on session – Scenario E
- Optional – Scenario I

**Group 5**

- First hands-on session – Scenario E
- Second hands-on session – Scenario A
- Optional – Scenario J

## Scenarios List

**Scenario A**

**Visualize this public space with provided tools:**

- Public space typology: Park
- Amenities: Sitting space, green space
- Location: Less dense suburban Montreal

**Scenario B**

**Visualize this public space with provided tools:**

- Public space typology: Pedestrian promenades
- Amenities: Safe streets, community engagement spaces, green spaces
- Location: Historical neighborhood in Montreal

**Scenario C**

**Visualize this public space with provided tools:**

- Public space typology: Street space

- Amenities: All ages-, all genders-, all abilities-, all identities-friendly environments

- Location: Residential neighborhood in Montreal

**Scenario D**

**Visualize this public space with provided tools:**

- Public space typology: Downtown plaza

- Amenities: Meeting spaces, sitting area, rest areas, versatile use space

- Location: Downtown Montreal

**Scenario E**

**Visualize this public space with provided tools:**

- Public space typology: Park

- Amenities: Rest areas, community engagement spaces, waterfront area

- Location: Dense urban area in Montreal

**Optional Scenarios**

**Scenario F**

**Visualize this urban space:**

- Public space typology: Urban garden

- Amenities: Educational programs, community gardening spaces

- Location: Near universities and colleges in Montreal

**Scenario G**

**Visualize this urban environment:**

- Public space typology: Waterfront sidewalk

- Amenities: Outdoor cafes, art installations, pedestrian paths, bike lanes

- Location: Along a river or lake in a mixed-use (residential and commercial uses) area of Montreal

**Scenario H**

**Visualize this community space:**

- Public space typology: Neighborhood square

- Amenities: Playgrounds, outdoor fitness equipment, community noticeboards, seasonal markets

- Location: Residential area in Montreal, possibly near schools and local businesses

**Scenario I**

**Visualize this communal area:**

- Public space typology: Transit plaza

- Amenities: Sheltered seating, transit information displays, public art, bike-share stations

- Location: Key transit hub in Montreal, adjacent to a metro station or major bus interchange

**Scenario J**

**Visualize this urban setting:**

- Public space typology: Alleyway

- Amenities: Street murals, pedestrian lighting, small business kiosks

- Location: Back alleys commercial district of Montreal

## F. Experiment Details

We fine-tuned Stable Diffusion XL using Direct Preference Optimization (DPO; Rafailov et al. 2024), closely following the original hyperparameters:

- **Batch Size:** 64

- **Learning Rate:** $1 \times 10^{-8}$ with 20% linear warmup

- **Beta** ($\beta$)**:** 5,000

- **Training Steps:** 500 for smaller subsets; 1,500 when combining the entire preference dataset

- **Hardware:** Single NVIDIA A100 80GB GPU

All preference values (continuous slider results) were discretized to binary labels (preferred vs. not preferred) for DPO compatibility. While a majority-voting procedure resolved multi-criteria conflicts, future work could explore methods that retain richer preference signals.

## G. Qualitative Observations

Figures 13, 14, and 15 compare baseline SDXL images against versions finetuned on specific or multiple LIVS criteria. While DPO alignment often improves features such as smooth surfaces or clearer paths, certain elements (e.g., ramps, multilingual signage) appear inconsistently, indicating the model's limited capacity for rendering specialized details.

## H. Additional Analysis: Intersectional Scoring

### H.1. Case Study IV: Scoring Patterns of SDXL Images by Intersectional Identities

Figures 16 and 17 plot average raw scores from intersectional groups (e.g., disability status × race/ethnicity) for SDXL-generated images across the six LIVS criteria. Participants with mobility challenges generally assigned lower *Accessibility* and *Safety* scores, highlighting the importance of soliciting localized, intersectional feedback in T2I alignment for public-space design.

*Prompt: A bike path separated from vehicular traffic by barriers.*

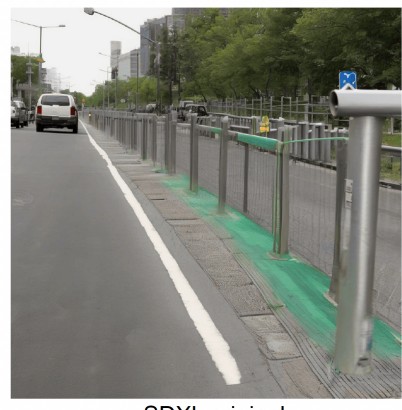 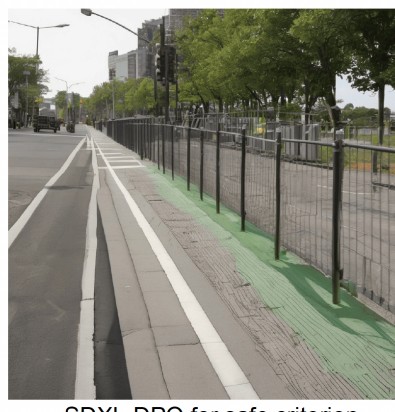 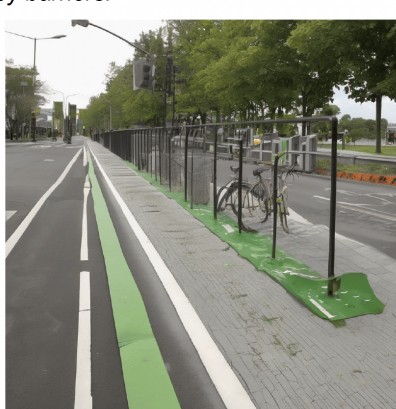

| SDXL original | SDXL-DPO for safe criterion | SDXL-DPO for all six criteria |

*Figure 13.* **Bike Path Scenario Focused on Safety.** From left to right: baseline SDXL output with minimal barriers; a safety-tuned version with stronger separation but lacking comfort features; and a multi-criteria DPO version featuring wider lanes and smoother transitions, though certain amenities remain missing.

*Prompt: A shopping mall with wide aisles, ramps, and accessible restrooms.*

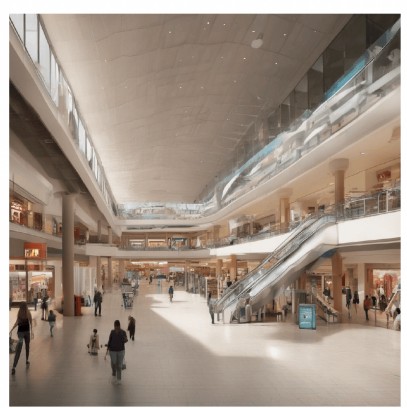 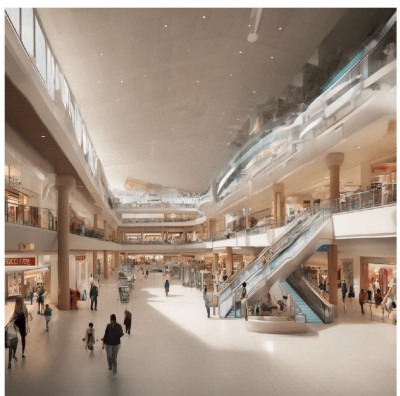 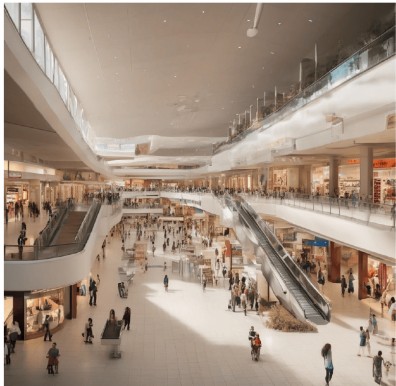

| SDXL original | SDXL-DPO for accessibility criterion | SDXL-DPO for all six criteria |

*Figure 14.* **Shopping Mall Scenario Emphasizing Accessibility.** From left: baseline SDXL with limited ramps; an accessibility-tuned version that adds more stairs; and a multi-criteria DPO output showing wider walkways but still struggling with ramp clarity.

*Prompt: A metro station decorated with artwork representing different heritages.*

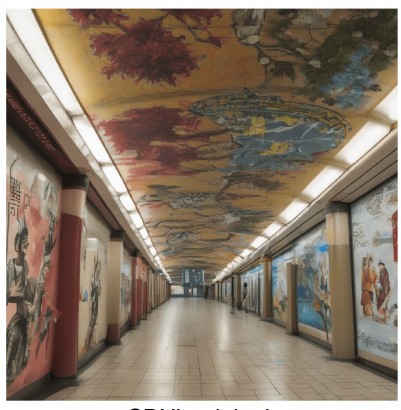 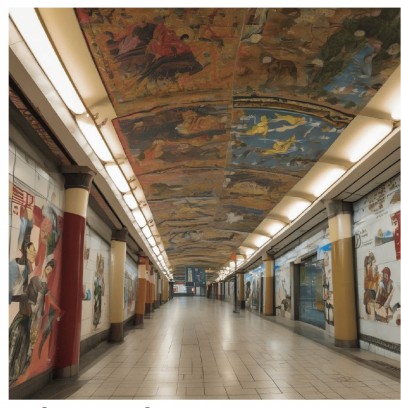 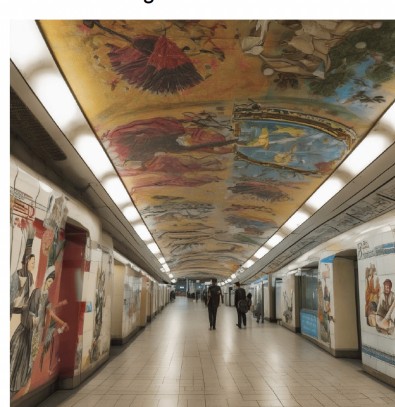

| SDXL original | SDXL-DPO for diversity criterion | SDXL-DPO for all six criteria |

*Figure 15.* **Metro Station Scenario Emphasizing Diversity.** From left: baseline SDXL with limited demographic variety; a diversity-focused model offering modestly varied individuals; and the DPO-tuned version showing slightly broader representation, though cultural markers (e.g., multilingual signs) remain muted.

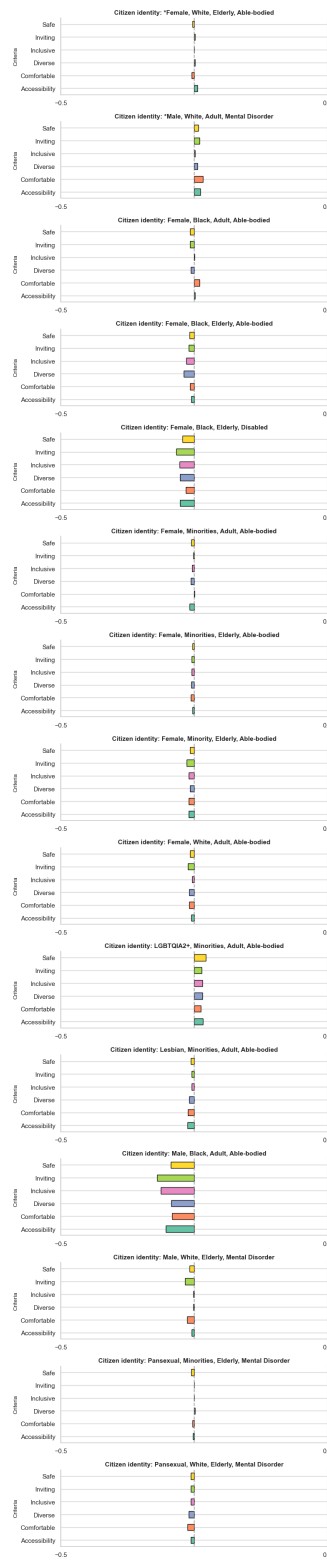

*Figure 16.* Main dataset: SDXL-scored public-space images, grouped by intersectional identity. Participants with mobility constraints often assigned lower *Accessibility* ratings.

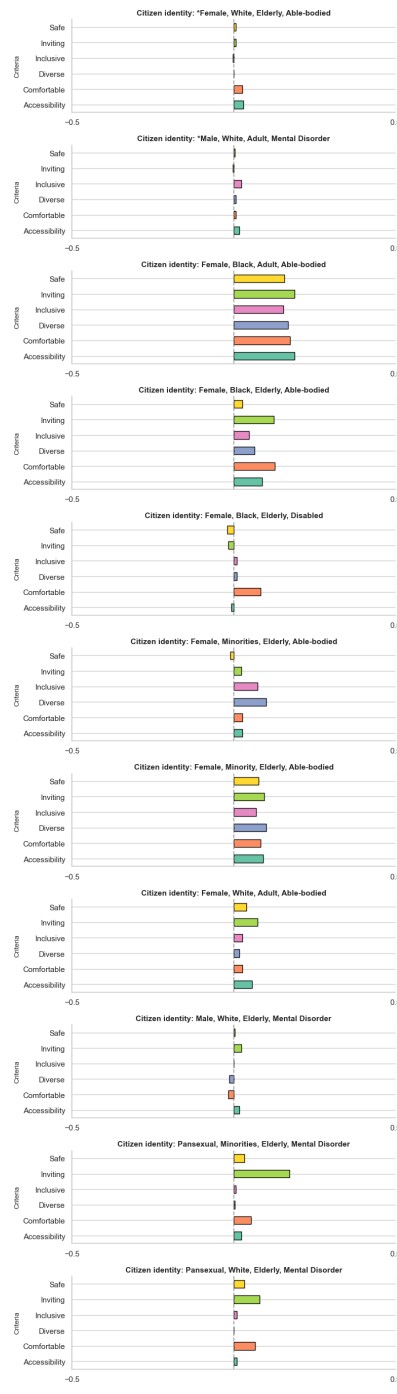

*Figure 17.* Evaluation dataset: SDXL-scored images, again grouped by intersectional identity. Patterns parallel those observed in the main dataset, with lower *Accessibility* or *Safety* scores among some groups.

# I. Variance

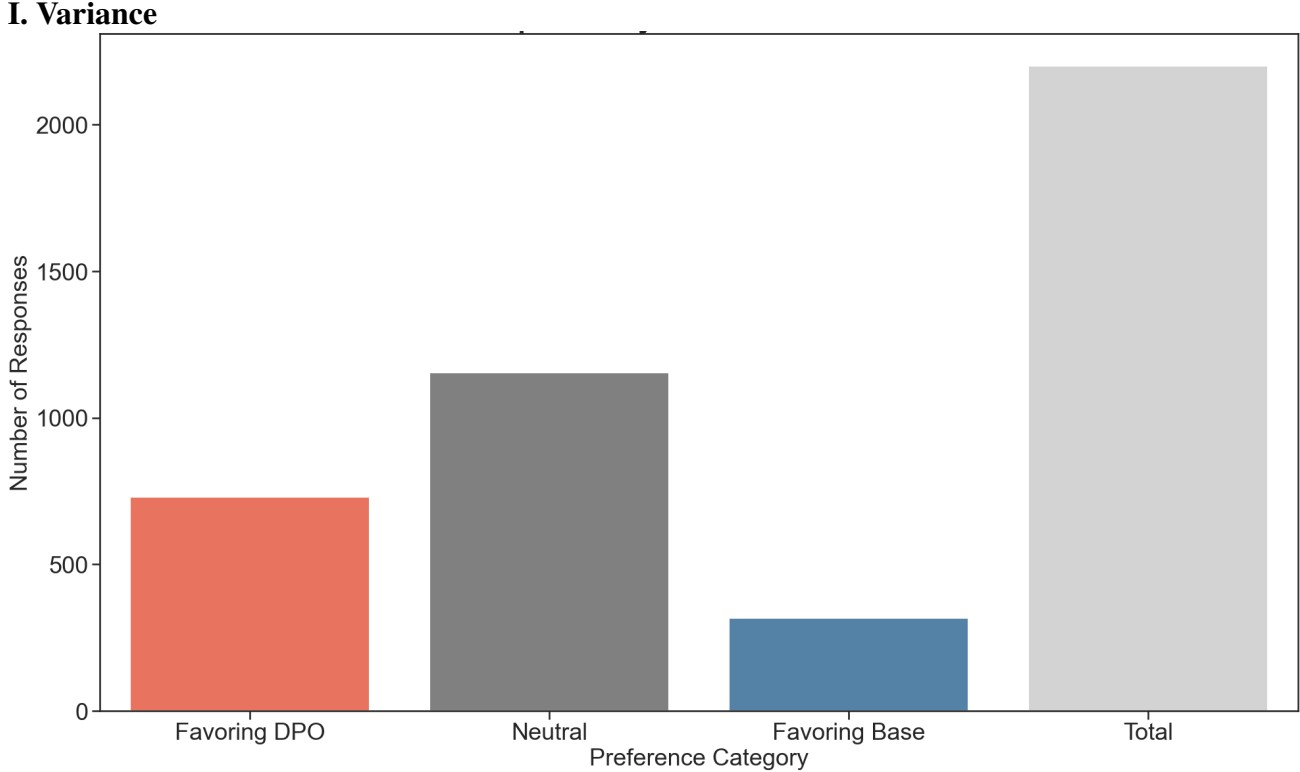

*Figure 18.* **Preference Distribution for DPO vs. Baseline.** Each bar represents the share of 2,200 final annotations favoring the DPO-aligned model, the baseline, or indicating neutrality. Neutral selections suggest subtle differences or partial fulfillment of criteria by both images.

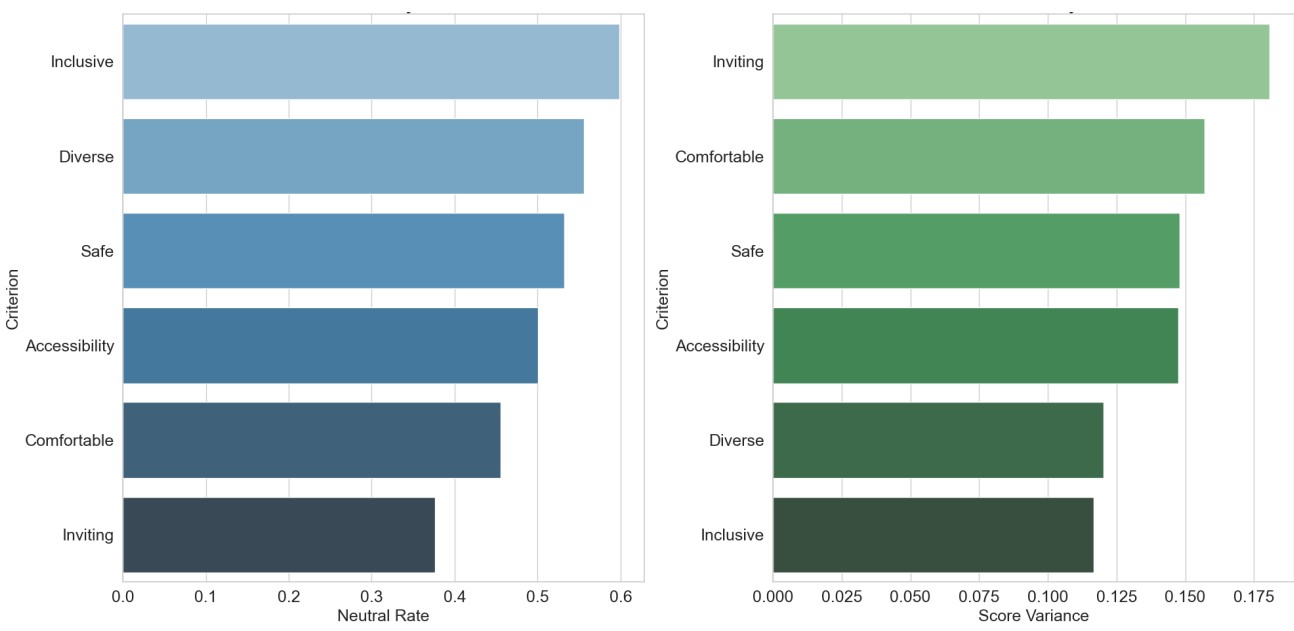

*Figure 19.* Main dataset: neutral ratings and variance by criterion. Higher variance in some criteria (e.g., *Inclusivity*) may reflect subjective or difficult-to-visualize concepts.

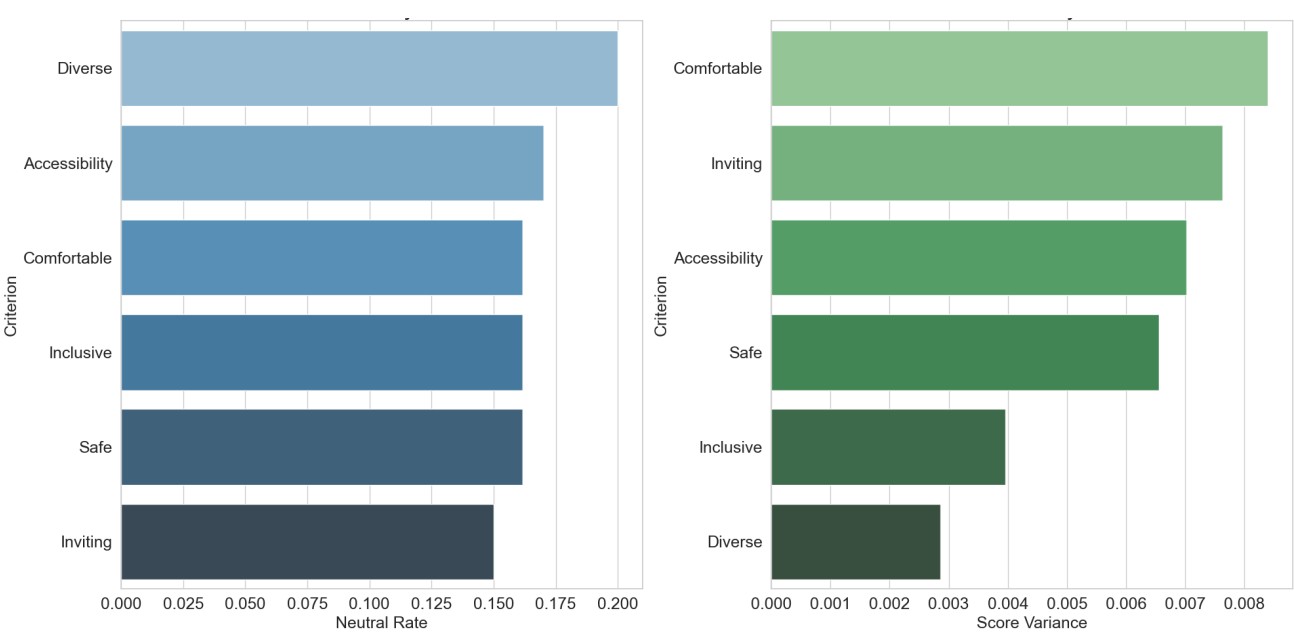

*Figure 20.* Evaluation set: neutral rating and variance by criterion. Similar patterns of higher neutrality persist for *Inclusivity* and *Diversity*.

