# OpenReview forum: "LIVS: A Pluralistic Alignment Dataset for Inclusive Public Spaces"
_ICML.cc/2025/Conference — ICML 2025 poster_

### Official Review · Reviewer_iv23 · 2025-03-12

**Overall Recommendation:** 2

**Summary:**

The paper presents the Local Intersectional Visual Spaces (LIVS) dataset, a community-driven benchmark designed to align text-to-image (T2I) models with intersectional criteria for inclusive urban design. Through a two-year collaboration involving 30 community organizations, the authors iteratively refined 634 initial design concepts into six core criteria (Accessibility, Safety, Comfort, Invitingness, Inclusivity, Diversity) using participatory workshops and 37,710 pairwise comparisons. By applying Direct Preference Optimization (DPO) to fine-tune Stable Diffusion XL (SDXL), they demonstrate improved alignment with these criteria.

**Claims And Evidence:**

Supported Claim: DPO enhances alignment for criteria with sufficient annotation support, as evidenced by the user study in Case Study I and Figures 6, 12–14.

Unsupported Claim: While the authors assert that their work proposes a pluralistic alignment framework for T2I models, this claim lacks justification. Since DPO's training signals remain confined to binary preference pairs and do not explicitly model intersectional interactions between criteria, no true multi-criteria framework is established. Notably, neither algorithmic adaptations nor multi-objective optimization strategies are introduced.

**Essential References Not Discussed:**

None

**Experimental Designs Or Analyses:**

Experiments center on four user study-based case analyses: Case Study I demonstrates DPO's effectiveness for high-data criteria; identity-driven analyses (Cases II/IV) reveal preference variations across demographics; Case III underscores the critical role of prompt design in evaluation reliability. A reliance on qualitative outcomes persists, with quantitative metrics omitted.

**Methods And Evaluation Criteria:**

The participatory methodology for dataset construction (e.g., workshops, iterative concept refinement, pairwise comparisons) is clearly articulated and contextually appropriate for capturing community-driven priorities. However, the technical alignment approach relies entirely on off-the-shelf DPO without innovations addressing multi-criteria challenges. Evaluations include user studies and qualitative analyses, but standard quantitative metrics for T2I alignment (e.g., CLIP scores, FID) are notably absent.

**Other Comments Or Suggestions:**

The paper’s main contribution is its dataset construction, which is well explained and offers useful insights for future work. However, the claimed pluralistic alignment framework is unclear, as the DPO algorithm remains binary and only the annotations are multi-criteria—a concept already explored in prior works (e.g., ImageReward). Overall, this work is more of an excellent project than a novel research contribution. I am willing to raise my score if the authors can address my concerns.

**Other Strengths And Weaknesses:**

None.

**Questions For Authors:**

Figure 1's caption references the inclusion of age, gender, race/ethnicity, and disability demographics, yet only age distributions are visually presented. Where are other demographic breakdowns (e.g., gender ratios) reported or illustrated?

**Relation To Broader Scientific Literature:**

This work is related to T2I alignment methods relying on global metrics (ImageReward, HPS) and multi-criteria preference learning (MPS). It extends these by grounding multi-attribute annotations and local urban design contexts.

**Theoretical Claims:**

No theoretical claims or formal proofs are presented.

---

> ### Author Rebuttal · Authors · 2025-04-01
>
> We greatly appreciate the reviewer’s feedback. In response, we have clarified our methodological framing of pluralistic alignment, explicitly acknowledged the limitations regarding global metrics such as CLIP and FID, and revised Figure 1 for consistency and clarity.
>
> ---
> ### 1: The claimed pluralistic alignment lacks support, relying on binary DPO without modeling intersectionality.
>
> **Response:**
> We propose a pluralistic alignment framework, as our dataset and workflow explicitly incorporate multiple criteria—Accessibility, Safety, Comfort, Invitingness, Inclusivity, and Diversity—each annotated independently. While we apply DPO, our primary aim is to test its capacity to capture community-driven, intersecting needs. The core contribution of this work lies not only in the dataset itself but in the participatory framework through which it was created. Our originality lies in operationalizing pluralistic alignment: from co-developing multi-criteria definitions with communities, to collecting fine-grained preference data, to empirically fine-tuning a T2I model. We further demonstrate how DPO can be extended to reflect localized values. Models fine-tuned on LIVS and other datasets developed using this framework have potential applications in community consultations and urban design processes. We hope the reviewer recognizes that this approach meaningfully advances alignment research by grounding it in real-world civic contexts. Our human preference data offer greater relevance for intersectional, community-based alignment than generic image quality metrics. The high proportion of neutral judgments, in particular, underscores the complexity of socio-spatial values—dimensions that standard T2I evaluation metrics are ill-equipped to measure.
>
> To further clarify this distinction, we have revised the relevant text in the Introduction section as follows.
>
> **Revised text in Introduction:**
> *“We propose a participatory data-collection framework that captures intersectional, multi-criteria feedback for T2I models in inclusive public-space contexts. While we refer to this as a *pluralistic alignment* approach to emphasize the local diversity of preferences, our method currently employs standard DPO with binary preference pairs, rather than a specialized multi-objective optimization algorithm. By integrating multiple locally defined criteria into preference annotations, we aim to expose both the potential and the limitations of a single-objective approach in accommodating intersectional needs.”*
>
> ---
> ### 2: No CLIP or FID; relies on user qualitative analysis
>
> **Response:**
> We intentionally foreground community-derived judgments over global image-similarity metrics, as our primary aim is to assess how well the generated images align with local, context-specific criteria. FID, measures similarity to a global real-image distribution and is not meaningful in our context. We also avoid CLIP-based metrics, as prior work ([1], [2]) shows weak correlation with human judgments. Given our focus on subtle, value-driven distinctions across communities, relying on CLIP risks missing the very nuances we aim to evaluate.
>
> Nonetheless, we acknowledge that such metrics can provide useful insights into broader questions—such as identifying common features shared by inclusive spaces across different typologies.
>
> **To reflect CLIP and FID limitations, we added this to the Limitations section:**
>
> *“Our evaluation centers on human preference judgments rather than traditional metrics such as CLIP scores. While these metrics are useful for quantifying generative fidelity, they fall short in capturing nuanced, local, or intersectional considerations. ”*
>
> ---
> ### 3: Figure 1 problem
>
> **Response:**
> Revised Figure 1 for clarity and consistency.
>
> ---
> ### 4: More project than a novel research contribution.
>
> **Response:**
> We respectfully disagree. Our framework and dataset are novel on multiple levels: (i) they capture multi-criteria feedback from diverse communities, thereby reflecting nuanced, real-world norms; (ii) they empirically demonstrate how DPO performs on intersectional needs, exposing a key gap in current alignment methods; and (iii) they introduce a participatory data collection framework that systematically grounds alignment in heterogeneous civic contexts. Unlike ImageReward and other prior efforts that rely on homogeneous annotator pools, our pluralistic dataset and analysis illuminate both the potential and the limitations of single-objective alignment when values meaningfully differ across populations. This combination of empirical insights and community-based methodology advances alignment research beyond conventional quality metrics, underscoring its broader social implications.
>
> ---
>
> **References:**
>
> [1] Ku et.al. VIEScore: Towards Explainable Metrics for Conditional Image Synthesis Evaluation. ACL 2024
>
> [2] Ku et.al. ImagenHub: Standardizing the evaluation of conditional image generation models. ICLR 2024

---

### Official Review · Reviewer_ABdq · 2025-03-14

**Overall Recommendation:** 4

**Summary:**

This paper introduces a new dataset LIVS, which encodes community-generated plurastic preference data towards text-to-image for urban planning. This dataset is is built from data collected from 30 community organizations to develop a framework of 6 axes along which urban public space design can be evaluated. Based on this framework, the authors collect the dataset of 38k human preference annotations.

The training split of this data is used to perform DPO finetuning on Stable Diffusion XL, which is then tested on the validation split for analysis. The authors find that the trained model has a considerable win rate over the baseline but also that half of the matchups have a neutral outcome, which is presumed to illustrate the subjective and plural nature of the criteria.

**Claims And Evidence:**

The authors claim that DPO taking into account multi-criteria feedback improves image generation in their considered space of urban planning. This claim is supported by their evidence (e.g. 70% win rate among non-neutral judgments). However, the extent of the claim is made weaker by the dominance of neutral judgments, accounting for more than half of the results. Given that collecting more training data to strengthen the claim may be costly due to the rigor involved, it might have been helpful to see a more nuanced breakdown using the existing training data: for example, by showing relative effects on different axes of evaluation when prioritizing those axes during DPO training.

The authors also claim that the significance of neutral judgments is that they highlight where "preferences are balanced or where further refinement is needed to accommodate complex intersectional needs". However, it is possible that a large proportion of neutral judgments also highlight axes of evaluation which may not be picked up by the participants comparing images. Particularly from the proportions displayed in Figure 5, as well as the description of late-joining participants in S4.2, it is possible that some of the axes (Diverse, Inclusive, Safe) may be too difficult or impossible for people to discern from concept art images.

**Essential References Not Discussed:**

None that I am aware of.

**Experimental Designs Or Analyses:**

Overall, the experimental designs and analyses appear well thought out. The authors explore multiple aspects of their results, including the effects of identity, intersectional interactions, and prompt compositions.

**Methods And Evaluation Criteria:**

The authors follow a meticulous and community-involved process to develop their framework of evaluation along 6 axes, and to collect training and testing data to illustrate the application of T2I in the urban planning space. Overall, this set of evaluations and analyses are well-defined and executed.

However, from their observations, the authors suggest that "less involvement in the knowledge-exchange process can lead to different or less pronounced alignment perceptions". For the purposes of expanding the scope of the study and data, it might have been helpful to expand upon definitions of the chosen concepts to allow more participants to offer their input and preference feedback with lower involvement. For instance, in addition to the short descriptions in S3.2, it would have been helpful to have compiled example images that illustrate the concepts being tested.

**Other Comments Or Suggestions:**

Figure 1 is organized in an unintuitive way. The y axis shows age group, and it appears that all participants were assigned to one of four age groups, but these appear in no particular order. Why are the datapoints not sorted by age along the y axis?

**Other Strengths And Weaknesses:**

The paper presents the results of a very methodical and well-thought out approach to gathering real preference data from communities, targeting real issues deemed important by those communities. Overall, the paper is original and clear, though its significance is weakened at points due to the weakness of the overall results, which I address in other sections.

**Questions For Authors:**

The authors conclude that the reason for a large number of "neutral" annotations is the highly "subjective and plural nature" of determining alignment to people's criteria. However, to me, this appears underjustified (Q1 and 2).
1. At what point in the process, i.e. Workshops and Interviews, do you consider the relevance of the proposed axes of evaluation in concept art images? Is the determination of the 6 axes driven solely by holistic participant experiences and suggestions, or also by the plausibility of distinguishing those axes from an image?
2. Similarly, it is suggested that the 1100 neutral rankings during evaluation are due to pluralistic evaluation. Is this statement supported by other evidence, or is it possible that the models differ very little (in generation for particular prompts) or participants find it difficult to measure an image along particular axes?
3. From the example images and results, it appears that many of the generated images have only very subtle differences after DPO finetuning. Was the conditional guidance scale tuned during generation to ensure that annotators would be able to distinguish key differences between images from the same prompt?

**Relation To Broader Scientific Literature:**

The paper explores the role of T2I models in urban planning from a perspective of pluralistic values. While the paper does not contribute to the pluralistic theory itself, it uses it as a framework to develop a set of criteria with a human-centric method that may be useful in other scenarios as well. The work relates to a lot of prior work in finetuning T2I models for human preferences, but again, focuses on the local and community-centric nature of preferences, and uses them to present more convincing findings.

**Theoretical Claims:**

No theoretical claims.

---

> ### Author Rebuttal · Authors · 2025-04-01
>
> We thank the reviewer for their feedback. We clarified how criteria were defined, explained neutral judgments, revised Figure 1, and added detail on axis-specific outcomes and image generation settings.
>
> ---
>
> ### Question 1: Axes Relevance and Plausibility
>
> **Response:**
> It was both, since participants were looking at images during the focus groups. We revised our Criteria Consolidation section to clarify that the final six criteria emerged from community input on design and from probing whether each dimension could be assessed through images.
>
> ---
>
> ### Question 2: Neutral Judgments and Multi-Criteria Evaluation
>
> **Response:**
> We clarify that neutral judgments reflect both genuinely balanced preferences and difficulties in visually encoding certain criteria, especially Inclusivity and Diversity.
>
> **New text in first paragraph in "Neutral Annotations as a Signal":**
> *“Approximately half of the final evaluations were rated as neutral. While these outcomes may initially seem to indicate a lack of meaningful improvement, they could also stem from genuinely balanced preferences and participants’ difficulty in identifying visual cues for certain criteria (especially Inclusivity and Diversity). In several interviews, participants noted that subtle or symbolic elements did not always appear clearly. This indicates an opportunity to incorporate neutral signals explicitly into training and to explore methods that better visualize intangible attributes, such as inclusive design features, beyond purely aesthetic details.”*
>
> ---
>
> ### Question 3: Guidance Scale and Subtle Differences
>
> **Response:**
> Yes, we employed a moderate guidance scale during the image generation process. To ensure meaningful variation in outputs, we conducted pilot runs and fine-tuned hyperparameters such as seed, guidance scale, and steps. We also generated 20 images per prompt and used a greedy selection strategy, choosing the 4 most distinct images based on CLIP similarity scores, as detailed in Algorithm 1. This approach helped balance the risk of generating images with either overly subtle differences or excessively stylized divergences. We provide further clarification in Appendix (C.2. Image Generation).
>
> ---
>
> ### Comment 1: Figure 1 Organization
>
> **Response:**
> We have revised the figure (Reviewer iv23 also finds this problematic).
>
> ---
>
> ### Comment 2: More Nuanced Breakdown per Axis
>
> **Response:**
> We include additional discussion of axis-specific outcomes (see *New Paragraph below*), showing that some axes (Comfort, Invitingness) benefit more noticeably from DPO with more annotated examples.
>
> **New Paragraph in "4.1 Additional Observations":**
> *“We further analyzed alignment improvements on each criterion by correlating annotation counts with the DPO model’s win rate. The criteria that received more annotations (Comfort and Invitingness) exhibited stronger improvements, suggesting that denser feedback can refine criteria-specific features more effectively. In contrast, Inclusivity and Safety showed a higher proportion of neutral or mixed outcomes, possibly reflecting both fewer annotations and the inherent difficulty of visually conveying representational aspects through T2I alone.”*
>
> **Please see the link below, which contains a figure on the Average Tendency Toward DPO by Criterion**
> https://anonymous.4open.science/r/livs-6E96/average-tendency.png
>
> ---
>
> **We hope these revisions address the reviewer’s concerns and enhance the clarity of our manuscript. Thank you for the valuable insights.**

---

### Official Review · Reviewer_adw2 · 2025-03-17

**Overall Recommendation:** 4

**Summary:**

The authors contribute LIVS, a benchmark for aligning text-to-image (T2I) models with respect to multiple criteria (Accessibility, Safety, Comfort, Invitingness, Inclusivity, and Diversity) in the context of urban public space design. The benchmark was developed via two-year participatory process with 30 community organizations in Montreal. The authors use DPO with 35,510 multi-criteria community preference annotations to align a Stable Diffusion XL model and find that: (1) the resultant generations can be better aligned with the preferences, (2) there remain a significant amount of neural ratings of the generations (possibly due to the complexity of modeling intersectional preferences), and (3) larger-scale alignment can be more effective. The authors also study the effect of prompt variation on community ratings of generations (observing that human-authored prompts are better at eliciting decisive preferences than synthetic prompts), and find that preferences vary across identities.

## Update after rebuttal

The authors provided a detailed response and suggested beneficial revisions based on my comments. I have maintained my (already high) score.

**Claims And Evidence:**

- The claims are generally supported by clear and convincing evidence.

- Lines 76-77: The authors claim that their "approach applies multi-criteria preference learning," but they ultimately collapse the multi-criteria preference annotations into a single binary annotation via majority aggregation.

- Figure 3 does not provide clear evidence for the "comprehensiveness" of the prompt dataset (line 262).

**Essential References Not Discussed:**

I am not aware of any essential references that were not discussed.

**Experimental Designs Or Analyses:**

- To perform preference learning, the authors collapse the multi-criteria preference annotations into a single binary annotation via majority aggregation, which does not preserve differing intersectional preferences.

- The resultant generations from the aligned SDXL model are compared to generations from the baseline SDXL model on a held-out set of prompts, which is sound.

**Methods And Evaluation Criteria:**

- The authors facilitated an extensive participatory design process over two years with 30 community organizations, consisting of public education, 11 workshops, 34 interviews, and inclusive data collection. This methodology is excellent, as it positions community members as co-creators of LIVS, from criteria design to data annotation, and captures complex and diverse local preferences.

- The authors augment human-collected prompts with synthetic prompts generated using GPT-4o, but the synthetic prompts are not validated by humans (only validated automatically). The authors note in line 370 that the synthetic prompts likely lack "contextual specificity."

**Other Comments Or Suggestions:**

- Minor comment: The use of "democratizing" (line 43) requires further contextualization in the paper. In particular, the use of "democratizing" might be a bit misleading given that the paper does not discuss, e.g., democratic governance structures for T2I models in urban planning [1].

- The authors should expand on how their work relates to participatory action research.

[1] https://aclanthology.org/2024.emnlp-main.184/

**Other Strengths And Weaknesses:**

Strengths:
- The authors focus on aligning models with local community preferences around inclusive urban planning, thereby advancing pluralistic alignment.

- The authors transparently document their ethics and inclusivity considerations, e.g., compensating community members, utilizing an inclusive and accessible data annotation interface.

- The paper is clearly written and well-organized.

- The authors offer numerous directions for future work, e.g., leveraging neutral annotations as a signal, disentangling ratings for overlapping criteria.

Weaknesses:
- The authors do not explicitly leverage neutral annotations or disagreements in annotations across criteria during preference learning.

**Questions For Authors:**

- On average, how many criteria did a community member annotate per image pair?

- How may the definition of criteria like Inclusivity be refined to lead to more distinct preferences? Should the definition be more prescriptive or descriptive?

- In Figure 8, do the left and right images in a single pair come from the same or different models?

**Relation To Broader Scientific Literature:**

- The authors thoroughly explore the relationship between their work and existing work on the alignment of generative models, intersectionality, visual generative modeling for urban spaces, and multi-criteria preference learning.

- The authors go beyond much prior work on global/universal alignment by capturing local multi-criteria preference annotations, specifically in the context of urban public space design.

- The authors go beyond the common conceptualization of Intersectionality as merely overlapping social groups [1] and discuss how overlapping forms of marginalizations affect people's preferences about, e.g., accessibility (Section 2.2).

- The authors build on the tradition of meeting community design objectives in urban planning.

- The paper is similar to prior work on multi-criteria preference learning, but does so by aligning SDXL with respect to multiple criteria (Accessibility, Safety, Comfort, Invitingness, Inclusivity, and Diversity) in the context of urban design.

- Some citations may not directly support the authors' claims, e.g., [2] in lines 115-119. The authors should double-check that all their parenthetical citations are directly relevant.

[1] https://dl.acm.org/doi/10.1145/3600211.3604705

[2] https://dl.acm.org/doi/10.1145/3613904.3642703

**Theoretical Claims:**

The authors did not make any theoretical claims.

---

> ### Author Rebuttal · Authors · 2025-04-01
>
> We thank the reviewer for their detailed and constructive feedback. Below, we respond to each comment. Where revisions are needed, we provide the updated text.
>
> ---
>
> ### Comment 1: Multi-criteria preference learning but collapsing annotations, neutral annotations not leveraged; disagreements across criteria are collapsed
>
> **Response:**
>
> We acknowledge that our current method for DPO training reduces multi-criteria signals into a single reward label. While this dataset is inherently multi-criteria, this reduction reflects a pragmatic simplification for the initial phase of DPO training. However, this also underscores a broader limitation of DPO: it is not straightforward to perform multi-criteria alignment—particularly when preferences across criteria conflict or when alignment must be criteria-aware within the same model. We have clarified this in the revised text and emphasize that future work is needed to develop methods capable of capturing such partial, intersecting, or contradictory preferences in a more principled way.
>
> **Revision in 2.1. Alignment of Generative Models:**
> *“Although the LIVS dataset contains multi-criteria feedback, we initially collapse these signals into a single preference label for each pair during DPO. This step overlooks conflicting or nuanced assessments across different criteria. Future work is needed to explore approaches that account for intersections and disagreements without forcing a single binary label, thereby preserving the richness of multi-criteria preference data.”*
>
> ---
>
> ### Comment 2: Figure 3 does not clearly show "comprehensiveness" of prompts
>
> **Response:**
> We revised the text to clarify that the word cloud is a preliminary visualization of different concepts within prompts. We rely on Jensen–Shannon Divergence (JSD) scores and scenario-based coverage to demonstrate prompt diversity. The figure is meant only as an illustrative snapshot.
>
> ---
>
> ### Comment 3: Synthetic prompts not validated by humans
>
> **Response:**
> We acknowledge that synthetic prompts were primarily evaluated using automatic diversity checks (JSD). As noted in the limitations, this expands coverage but may lack the contextual specificity of human-authored prompts. Future work needs to incorporate human validation to improve relevance.
>
> ---
>
> ### Comment 4: Use of "democratizing" (line 43) requires more context
>
> **Response:**
> We removed the term “democratizing” and clarified that T2I tools aim to lower barriers to community participation in design. Thank you for the reference.
>
> **Revised Sentence in Introduction:**
> *“These developments can benefit communities by making design processes more accessible—enabling broader engagement among non-expert stakeholders in architecture, urban planning, and environmental visualization.”*
>
> ---
>
> ### Comment 5: Citation issue
>
> **Response:**
> Rectified.
>
> ---
>
> ### Comment 6: Participatory action research grounding
>
> **Response:**
> We added the following paragraph in the `Methodology: Building the LIVS Dataset` section.
>
> **Revised Paragraph:**
> *“Participatory Action Research (PAR). In line with the principles of PAR, our community-based approach centers on iterative, collaborative inquiry and reciprocal learning throughout the dataset development process (Israel et al., 1998; Cornish et al., 2023). By involving local organizations as active co-researchers, we ensured that the framing of inclusion, safety, and other design criteria emerged from lived experiences rather than external prescriptions. This iterative feedback loop aligns with PAR’s emphasis on collective problem-solving and empowerment, as participants guided each stage of data collection and model evaluation while gaining familiarity with T2I technology and its potential applications in urban contexts.”*
>
> ---
>
> ### Comment 7: Question—On average, how many criteria per image pair?
>
> **Response:**
> On average, each image pair received approximately 1.49 non-zero (decisive) annotations from community members.
>
> ---
>
> ### Comment 8: Question—Refining definitions like "Inclusivity" for more distinct preferences
>
> **Response:**
> This is a trade-off: while detailed definitions may yield clearer preferences, our prompts were prescriptive and annotations observational to avoid biasing participants. Now, with the full dataset, we plan to analyze each criterion by identifying subdimensions and associated objects for emerging patterns.
>
> ---
>
> ### Comment 9: Question—Figure 8: Do left and right images come from the same or different models?
>
> **Response:**
> Regarding Figure 9 (since Figure 8 pertains to prompt methods), both images were generated by the same SDXL model.
>
> ---
> **We appreciate the suggestions, which helped clarify multi-criteria signals, prompt diversity, and neutral annotations. We believe the revisions address the reviewer’s concerns and strengthen the paper.**
>
> **References**
> 1. Israel, B. A. et al. (1998). Annu. Rev. Public Health
> 2. Cornish, F. et al. (2023). Nat. Rev. Methods Primers

---

> > ### Comment · Reviewer_adw2 · 2025-04-03
> >
> > Thank you for the detailed response and revisions based on my comments! I would like to maintain my (already high) score.

---

### Official Review · Reviewer_8G5x · 2025-03-19

**Overall Recommendation:** 3

**Summary:**

The authors of this paper collected a human preference dataset (called LIVS) of generated images about public spaces. The preference focuses on evaluating six metrics, including Accessibility, Safety, Comfort, Invitingness, Inclusivity, and Diversity. Then, they finetune a Stable Diffusion XL model using Direct Preference Optimization (DPO). The finetuned SDXL model's generated images got more preference than the original SDXL's images (32% vs 14%).

**Claims And Evidence:**

- The authors claim that the LIVS dataset captures diverse, community generated dimensions of inclusive public space design. I think this claim is clearly supported by the data collection process, where communities participated in the workshops and condensed the evaluation metrics into six aspects.
- The authors showed the effectiveness of using the collected LIVS dataset (via DPO) to finetune SDXL towards human preference aligned with the six metrics. This claim is supported by the human evaluation experiment to compare the finetuned SDXL and the original one (Section 4.1).
- The authors suggested that their results show the influence of participant identities on model preferences and the difference of the generated images resulted from human-authored and AI-generated prompts. These claims are supported by Figure 7 and Figure 8, respetively.

**Essential References Not Discussed:**

No.

**Experimental Designs Or Analyses:**

The experimental analyses are mainly based on statistical analysis (Figures 4, 5, 7, 8), which show the distributions of ratings. This is the correct way to do the analysis.

**Methods And Evaluation Criteria:**

The data collection is professional, where detailed instructions were given in a series of workshops and the annotators had sufficient understanding of the topic about public space design. The DPO is properly used to finetune SDXL with the collected preference data. Finally, it is correct to use human evaluation to check if the finetuned SDXL achieves better results than the original SDXL.

**Other Comments Or Suggestions:**

No.

**Other Strengths And Weaknesses:**

I want to raise a fundamental question regarding the necessity of developing the LIVS dataset to improve the image generation models' capacity in generating images that align with the six aspects in designing public spaces. I think it might improve the images better by specifying a better, more detailed prompt. A simple prompting method is by appending the six criteria to the prompts. An example can be "a shopping mall with wide aisles, ramps, and accessible restrooms. It should follow these criteria: Accessibility, Safety, Comfort, Invitingness, Inclusivity, and Diversity." How does this strategy improve the generated images?

**Questions For Authors:**

Why were only three randomly selected criteria from the total of six criteria shown in each annotation? Why not use all six criteria? Since the annotators have taken time to check the images, it would not take much more time to finish the evaluation on the remaining three criteria. Evaluating on all six criteria could results in more data with high efficiency.

**Relation To Broader Scientific Literature:**

The collected LIVS dataset would be useful in the community conerning public space design.

**Theoretical Claims:**

N/A

---

> ### Author Rebuttal · Authors · 2025-04-01
>
> Thank you for the constructive feedback. We clarified the rationale behind using community-informed prompts over universal keywords, explained the three-criteria annotation design, and updated key sections for clarity.
>
> ---
> ### Comment 1: Necessity of the LIVS Dataset vs. Enhanced Prompting
> **Response:**
> We recognize that refined prompting strategies—for instance, appending keywords such as *accessibility*, *safety*, or *inclusivity*—can guide generative models toward more targeted outputs. Approximately half of the prompts used to generate images included at least one criterion (see Table 1). However, LIVS emphasizes that *lived experience* varies considerably across social identities and contexts. In many urban settings, local communities articulate inclusivity or accessibility in ways specific to their sociocultural histories. As demonstrated in prior research (e.g., Beebeejaun, 2017; McAndrews et al., 2023), relying solely on universal keywords can impose a fixed and potentially biased interpretation of inclusive space. By contrast, our dataset incorporates diverse local knowledge and community priorities, reducing the risk of reproducing a single design standard that may neglect intersectional needs (Crenshaw, 1997; Low, 2020).
>
> **Table 1: Prompts with at least one of the six criteria vs. prompts without explicit criteria, and corresponding neutral response performance.**
>
> | Prompt Type      | % of Prompts | % Neutral Responses |
> |------------------|--------------|----------------------|
> | Without Criteria | 51.67%       | 21.76%               |
> | With Criteria    | 48.33%       | 26.26%               |
>
> We adopted a localized approach without imposing a predefined notion of inclusive space on participants. Our objective was to maximize the diversity of both images and prompts, allowing for an emergent understanding of what *inclusive* or *diverse* space could mean in different contexts (Anttiroiko & De Jong, 2020; Madanipour, 2010). To our understanding, this design encourages community-driven insights rather than a one-size-fits-all approach to inclusivity.
>
> **Revised Paragraph (in *Prompting and Early Feedback*):**
> *“Although refined prompting techniques can shape generative outputs, universal keywords alone may overlook local sociocultural and historical contexts (Anttiroiko & De Jong, 2020; Beebeejaun, 2017; Talen, 2012). Our goal in creating the LIVS dataset was to integrate granular, community-generated perspectives on accessibility, safety, and inclusivity. By embedding localized knowledge, we reduce the likelihood of producing a uniform design standard that might disregard certain intersectional needs (Low, 2020; Madanipour, 2010; McAndrews et al., 2023).”*
>
> ---
>
> ### Comment 2: Annotating Only Three of the Six Criteria per Comparison
>
> **Response:**
> During our tutorial workshop, participants indicated that rating fewer or more than three criteria simultaneously was cognitively demanding and reduced their focus on each image. Including more criteria also gave a sense of reduced progress, especially since the initial annotation batches comprised around 1,000 comparisons, later reduced to 750. We also observed that displaying six rating elements constrained the interface, limiting visibility of the image pairs and making it harder to assess spatial details. As a result, we randomly assigned three criteria per annotation task. This approach balanced coverage and participant effort while preserving image clarity. We confirmed that all six criteria received substantial coverage across multiple batches.
>
> **Revised Paragraph (in *Annotations and Evaluation*):**
> *“To optimize data quality and minimize cognitive fatigue, we randomly presented three of the six criteria in each comparison. During pilot trials, participants found evaluating all six criteria difficult, which reduced their sense of progress and visual engagement. The multiple rating elements also constrained image size, making it harder to assess spatial details. Focusing on three criteria enabled more meaningful engagement. Over successive annotation batches, we ensured that all six criteria were robustly evaluated.”*
>
> ---
>
> **We hope these clarifications and revisions address Reviewer's concerns. We appreciate their detailed comments, which have helped strengthen the methodological clarity and contextual framing of the paper.**
>
> ---
>
> **References**
> 1. Anttiroiko, A.-V., & De Jong, M. (2020). *The Inclusive City*. Springer.
> 2. Beebeejaun, Y. (2017). Gender, urban space, and everyday life. *J. Urban Affairs, 39*(3).
> 3. Crenshaw, K. (1997). Demarginalizing intersectionality. In *Feminist Legal Theories*. Routledge.
> 4. Low, S. (2020). Social justice and public space. In *Companion to Public Space*. Routledge.
> 5. Madanipour, A. (2010). *Whose Public Space?* Routledge.
> 6. McAndrews, C. et al. (2023). Toward gender-inclusive streets. *J. Planning Lit., 38*(1).
> 7. Talen (2012). *Design for Diversity*. Routledge.

---

### Decision · Program_Chairs · 2025-05-01

**Decision:**

Accept (poster)

**Comment:**

Reviewers are generally (though not unanimously) very positive about this paper, which presents a dataset for urban space design generated in a 2-year participatory design process with 30 community organizations. In authors’ words from the rebuttal, they “…apply DPO …to test its capacity to capture community-driven, intersecting needs. The core contribution of this work lies not only in the dataset itself but in the participatory framework through which it was created. Our originality lies in operationalizing pluralistic alignment: from co-developing multi-criteria definitions with communities, to collecting fine-grained preference data, to empirically fine-tuning a T2I model”

A few suggestions came up during rebuttal that I thought would be worth further consideration in the final version:
- recommendation to add quantitative image similarity analysis, possibly just to show that it does not align with community preferences
- while the rebuttal point that this approach maximizes diversity over a single strategy of adding keywords is well taken, more discussion to clarify this point in the final version may help